# PIP$_2$-mediated oligomerization of the endosomal sodium/proton exchanger NHE9

Surabhi Kokane[1,4], Ashutosh Gulati [1,4], Pascal F. Meier[1,4], Rei Matsuoka[1,4], Tanadet Pipatpolkai [2], Giuseppe Albano[3], Tin Manh Ho [3], Lucie Delemotte [2], Daniel Fuster [3] ✉ & David Drew [1] ✉

The strict exchange of Na$^+$ for H$^+$ ions across cell membranes is a reaction carried out in almost every cell. Na$^+$/H$^+$ exchangers that perform this task are physiological homodimers, and whilst the ion transporting domain is highly conserved, their dimerization differs. The Na$^+$/H$^+$ exchanger NhaA from *Escherichia coli* has a weak dimerization interface mediated by a β-hairpin domain and with dimer retention dependent on cardiolipin. Similarly, organellar Na$^+$/H$^+$ exchangers NHE6, NHE7 and NHE9 also contain β-hairpin domains and recent analysis of *Equus caballus* NHE9 indicated PIP$_2$ lipids could bind at the dimer interface. However, structural validation of the predicted lipid-mediated oligomerization has been lacking. Here, we report cryo-EM structures of *E. coli* NhaA and *E. caballus* NHE9 in complex with cardiolipin and phosphatidylinositol-3,5-bisphosphate PI(3,5)P$_2$ lipids binding at their respective dimer interfaces. We further show how the endosomal specific PI(3,5)P$_2$ lipid stabilizes the NHE9 homodimer and enhances transport activity. Indeed, we show that NHE9 is active in endosomes, but not at the plasma membrane where the PI(3,5)P$_2$ lipid is absent. Thus, specific lipids can regulate Na$^+$/H$^+$ exchange activity by stabilizing dimerization in response to either cell specific cues or upon trafficking to their correct membrane location.

Na$^+$/H$^+$ exchangers facilitate the exchange of monovalent cations (Na$^+$/Li$^+$/K$^+$) and protons (H$^+$) across membranes to regulate intracellular pH, sodium levels and cell volume[1]. In bacteria these ancient proteins work to either generate a sodium-motive-force or to alleviate sodium-salt stress[2,3]. In mammals, there are 13 different Na$^+$/H$^+$ exchangers (NHE) belonging to the SLC9A (NHE1-9), SLC9B (NHA1-2) or sperm specific SLC9C families[1,4]. NHEs differ in their substrate preferences, kinetics, subcellular and tissue distribution[1]. Isoforms NHE1-5 are primarily expressed on the plasma membrane and have important physiological roles linked to cellular pH homeostasis[1]. Isoforms NHE6-9 are primary localized to intracellular compartments and work in concert with the V-type ATPase to fine-tune pH in their respective organelles[5,6].

Na$^+$/H$^+$ exchangers are physiological homodimers belonging to the "NhaA-fold", so-named after the first fold-representative to be determined[2,7,8]. The transporter is made up of two distinct domains, a dimerization domain and an ion-transporting (core) domain. The 6-TM core domain undergoes global, elevator-like structural transitions to translocate ions across the membrane against the anchored dimerization domain[9,10]. Cryo-EM structures of SLC9A, SLC9B and SLC9C have been determined and show a similar architecture to bacterial homologous structures with 13 transmembrane (TM) helices[11–16]. Folds of secondary-active transporters are nearly always made up from two copies of a conserved repeat unit, related by internal structural-symmetry[17]. Interestingly, the structural-inverted repeats in proteins

[1]Department of Biochemistry and Biophysics, Science for Life laboratory, Stockholm University, Stockholm, Sweden. [2]Department of Applied Physics, Science for Life Laboratory, KTH Royal Institute of Technology, Stockholm, Sweden. [3]Department of Nephrology and Hypertension, Inselspital, Bern University Hospital, University of Bern, Bern, Switzerland. [4]These authors contributed equally: Surabhi Kokane, Ashutosh Gulati, Pascal F. Meier, Rei Matsuoka. ✉e-mail: daniel.fuster@unibe.ch; ddrew@dbb.su.se

with the NhaA-fold seem to be the only transporter members where the structural-repeat unit varies between its members. NhaA is made up of 12 TMs and has a 5-TM structural-inverted repeat, SLC9A and SLC9C are made up of 13-TMs and have a 6-TM structural-inverted repeat, whereas SLC9B has 14-TMs and a 7-TM structural-inverted repeat[15,16]. Although the 6-TM core ion translocation domain is conserved, the varying number of helices alters the structure of the dimerization interface. It seems that there has been evolutionary pressure to fine-tune oligomerization in the NhaA-fold for a functional reason.

Previous studies using native MS, MD simulations and thermal-shift assays, identified that NhaA oligomerization is dependent on cardiolipin[18–21]. NhaA has a weak dimer interface mediated by a β-hairpin domain, which may make dimerization propensity and activity more lipid-dependent than of other Na⁺/H⁺ exchangers[19,20,22]. Consistent with this reasoning, dimerization of the bacterial homologue NapA lacking β-hairpins and with a larger dimerization interface was found to be lipid independent[19]. It was further shown that NhaA was unable to compensate salt-stress in a *E. coli* strain deficient in cardiolipin[21]. Since cardiolipin biosynthesis in *E. coli* is increased upon salt-stress and, because NhaA is required to alleviate salt-stress[2,23], it was proposed that lipid binding might be a means to regulate in vivo NhaA activity[18]. Despite this attractive hypothesis, structural validation of cardiolipin binding to *E. coli* NhaA has been lacking.

In addition to NhaA, native mass-spectrometry and thermal-shift assays have also shown that the endosomal Na⁺/H⁺ exchanger NHE9 (SLC9A9) dimer co-purifies with negatively-charged lipids matching the mass of PIP₂[24]. NHE9 regulates the luminal pH of endosomes[6,25] and mutations in SLC9A9 have been associated with neurological disorders such as familial autism, ADHD and epilepsy[6,26]. Interestingly, although endosomal NHE9 has a larger dimer interface than NhaA, its size is ~30% smaller than in NHE1[27] (Supplementary Fig. 1a). The NHE9 protein has an additional ~60 residue extension of the TM2 -TM3 loop, which AlphaFold2[28] predicts forms a β-hairpin (Supplementary Fig. 1b, c). The TM2-TM3 loop is only present in organellar isoforms NHE6, NHE7 and NHE9 (Supplementary Fig. 1c). Similar to NhaA, multimer AlphaFold2[28] predicts the β-hairpins will interact with each other at the dimerization interface (Supplementary Fig. 1d). The TM2-TM3 loop domain was too dynamic to be modelled in previous cryo-EM structures of NHE9, yet its positioning suggested that it may also contribute to lipid binding (Supplementary Fig. 1e)[24]. Indeed, the mutation of two lysine residues K105 and K107 in the unresolved TM2-TM3 loop domain was sufficient to abolish PIP₂ stabilization, indicating this might be the case[24]. Moreover, the C-terminal tail of NHEs regulate ion-exchange activity, but the mechanism for activation is currently unclear[1,27,29]. In the previous NHE9 structure, the C-terminal tail could not be modelled, namely, residues 488 to 644[24].

Here, we set out to determine cryo-EM structures of the β-hairpin containing NhaA and NHE9 Na⁺/H⁺ exchangers to resolve the proposed lipid-mediated oligomerization and to establish how lipid-binding and the C-terminal tail of NHEs could regulate ion-exchange activity.

## Results

### Cryo-EM structure of the NhaA homodimer bound to CDL at pH 7.5

*E. coli* NhaA has become a model system for Na⁺/H⁺ exchangers[2,20]. Despite extensive biochemical, biophysical, and computational analysis, most structures of NhaA were crystallized as monomers[11,20]. The NhaA homodimer was observed in crystallo, but at 3.5 Å resolution it was unclear if lipids mediated dimerization[22]. To clarify the role of cardiolipin, a NhaA triple mutant A109T, Q277G, L296M retaining WT-like activity[22] was purified in detergent supplemented with cardiolipin (CDL) for structural investigation by cryo-EM. Following sample preparation optimization, a data set of 14,329 movies was collected and a cryo-EM map at 3.3 Å resolution was calculated, according to the gold-

standard Fourier shell correlation (FSC) 0.143 criterion (Supplementary Fig. 2a, Supplementary Table 1).

Similar to previous NhaA crystal structures, the cryo-EM structure of NhaA adopts the inward-facing conformation (Fig. 1a)[11,20]. Between inactive pH 4 and partially-active pH 6.5, a conformational switch termed a pH gate allows hydrated Na⁺ ions to enter the cavity and controls the pH at which NhaA becomes activated (Fig. 1a)[20,30]. As such, the mutation of charged residues on the cytoplasmic surface can alter the pH of NhaA activation and mutation of surface histidine residues to alanine abolished activity altogether[20,30]. Similar to the monomeric crystal structure at pH 6.5[20], an intracellular cavity in the cryo-EM structure at pH 7.5 has opened to enable Na⁺ accessibility to the ion-binding site (Fig. 1a).

The cryo-EM structure of NhaA is of the retained physiological homodimer (Fig. 1b). As expected from native MS and MD simulations[19], thermal-shift assays[18] and functional activity analysis[21,31] map density supported the modelling of cardiolipin at the dimerization interface (Fig. 1b, Supplementary Fig. 2b). One CDL lipid (CDL 1) was located in the middle of the positively-charged part of the dimerization interface, flanked by two additional CDL lipids (CDL 2) bound at the ends of the dimer interface (Fig. 1b, c). Although this part of the dimerization interface is very positively-charged with four arginine side chains per protomer contributing − R203, R204, R245 and R250 − only the symmetry-related R204 residues directly interacts with the phosphate headgroups of the central CDL 1 lipid (Fig. 1d). Additional hydrogen bonds are contributed by T205 and W258 to oxygen atoms of the CDL 1 phosphoester bonds (Fig. 1d). Deeper in the cleft, the side chain of W258 hydrogen bonds to R204 and also interacts with both CDL1 and CDL 2 lipids. The flanking CDL 2 lipids form hydrogen bonds to R204 and resides in a hydrophobic pocket lined by residues W258, Y261, and L262 from one protomer and L197, V207, and L210 from the other protomer (Fig. 1e).

The co-ordination of the central CDL 1 lipid seen in the structure of the NhaA triple mutant is entirely consistent with computational analysis of CDL binding sites summarized from the analysis of more than 40 different *E. coli* proteins[32]. Moreover, an R203A and R204A double-mutant abolished CDL-induced thermostabilization of purified NhaA[20]. The ion-binding site is located between cross-over helices in the core domain, yet accessibility to the ion-binding site is likely to be influenced by hydrophobic gating residues on the dimerization helix of TM2[15,33]. It seems that stabilization of the scaffold by CDL will favour Na⁺ accessibility to the ion-binding site, which would be consistent with the higher affinity for ²²Na⁺ as observed upon the addition of CDL to detergent-solubilised NhaA[31].

### Cryo-EM structure of NHE9* with TM2-TM3 β-hairpin loop domain at low pH 6.5

NHE9 was predicted by AlphaFold2[28] to harbour a β-hairpin loop domain between the topologically equivalent helices, TM2-TM3, to the β-hairpins in NhaA (Supplementary Fig. 1d, Supplementary Fig. 3a). In the AlphaFold2[28] monomer vs multimer prediction, the β-hairpins in NHE9 re-adjust their position to interact with the β-hairpin from the neighbouring protomer (Supplementary Fig. 3b). The previous modelled structure of *E. caballus* NHE9* (residues 21 to 488; out of 645) was determined by cryo-EM from protein purified in the detergent LMNG with CHS at pH 7.5, but the map density was too poor to model the loop domain[24] (Supplementary Fig. 1e). Here, we instead performed the collection of NHE9* at pH 6.5 where NHE9 is thought to be less active, and therefore presumably less dynamic. Overall, we collected ~4300 movies and the final 3D-reconstruction contained data of ~244,300 particles, from which an EM map could be reconstructed to 3.3 Å according to the gold-standard Fourier shell correlation (FSC) 0.143 criterion (Supplementary Fig. 4a). Comparison of the NHE9* structures determined at pH 7.5 and pH 6.5 showed no apparent differences (Supplementary Fig. 4b). We suspected that the respective

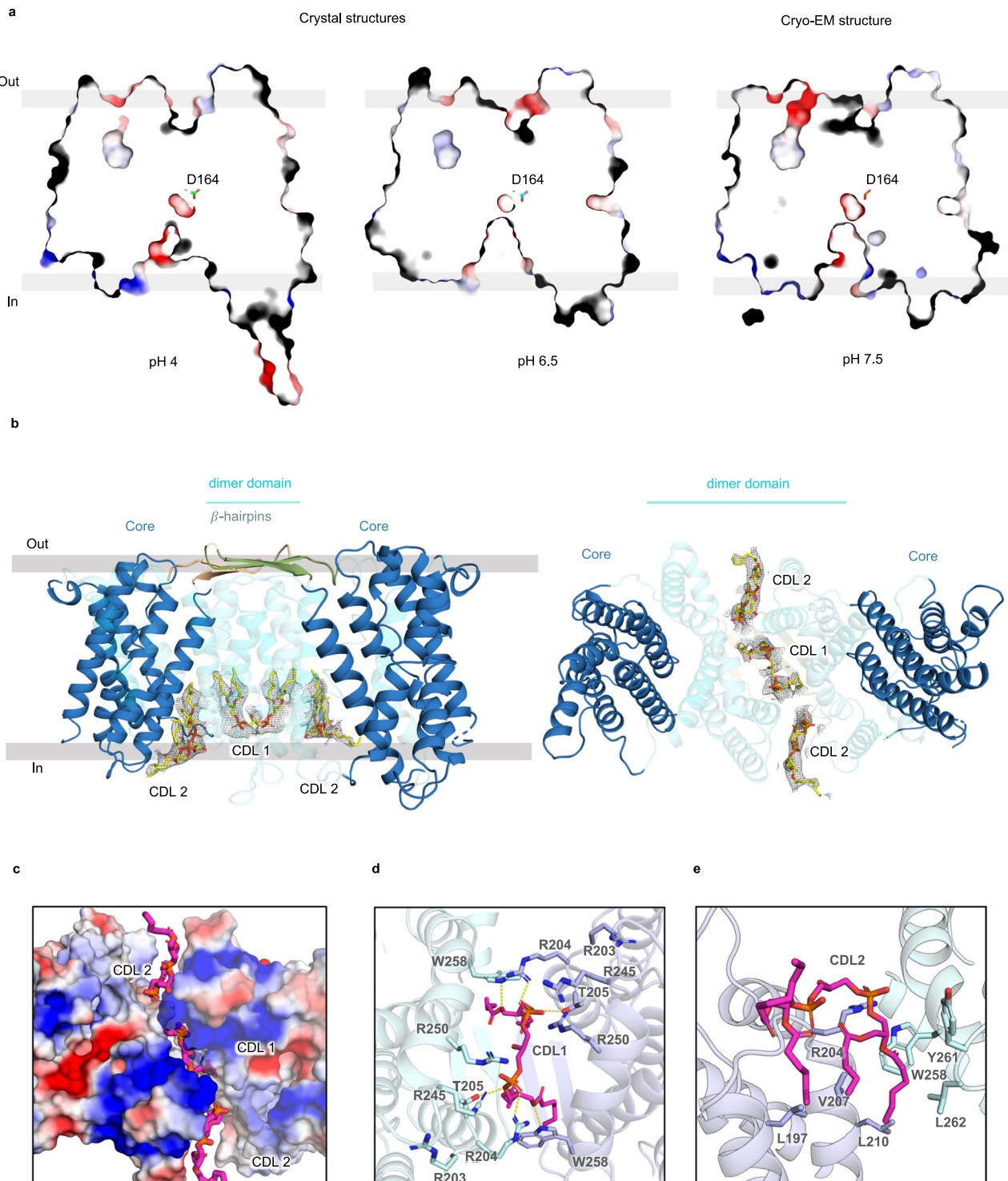

**Fig. 1 | Structure of cardiolipin bound to NhaA homodimer. a** Slice through an electrostatic surface representation of NhaA structures, perpendicular to the membrane plane, at inactive pH 4.0 (left, PDB ID: 4AU5), active pH 6.5 (right, PDB ID: 7S24) and pH 7.5-CDL bound. The ion-binding aspartate is indicated and shown as sticks (green, cyan and orange respectively). The ion-binding funnel at the cytoplasmic side is much more open at active pH 6.5 and pH 7.5-CDL bound structure than at pH 4. **b** *left:* Model of NhaA homodimer obtained by cryo-EM and with β-hairpin loop domains (palegreen, sand, cartoon). Large densities between the dimerization domains indicate the presence of three CDL lipids (yellow, sticks), which were modelled. *right* Cytoplasmic view of NhaA model with the three CDL lipids fitted between the dimer domains. **c** Cytoplasmic view of an electrostatic surface potential map of the respective monomers and the central CDL 1 and symmetry-related CDL 2 lipids (magenta, sticks) positioned between the hydrophobic and positively-charged dimerization interface (electrostatic charge illustrated from positive: blue, to negative: red). **d** Cytoplasmic view of the CDL binding interface of NhaA (cartoon, monomers colored slate and cyan respectively) with CDL 1 (magenta, sticks) and lipid-coordinating residues (sticks, colored per monomer) putative polar interactions indicated by dashed lines. **e** CDL2 binding interface of NhaA (cartoon, monomers colored slate and cyan respectively) with CDL 2 (magenta, sticks) and interacting residues in sticks (colored per monomer).

final 3D reconstructions may still contain heterogeneity and we therefore performed 3D-variability analysis in CryoSPARC[34], followed by clustering (principal component) analysis to separate further classes. Selecting 42% of the particles for heterogeneous refinement, enabled an improved cryo-EM map reconstruction of NHE9* at pH 6.5, with additional map features for the TM2-TM3 loop domain (Supplementary Fig. 5a). Further homogenous refinement applying C2 symmetry improved the cryo-EM density for the TM2-TM3 loop domain (Supplementary Fig. 5a), yet these features were not retained after masked refinement. For model building we therefore generated a composite map, by combining the maps before and after masked refinement, with an overall resolution estimate of 3.6 Å, although the final model resolution is worse at 4.0 Å resolution (Supplementary Fig. 5b, Supplementary Table 1). Subsequently, we refined the Alpha-Fold2 NHE9 multimer model into the NHE9* cryo-EM maps with some further manual building when needed (Methods and Fig. 2a).

At the extracellular end of TM2, a flexible loop formed by residues P72 to D82 were predicted by AlphaFold2[28] as extending towards the centre of the dimerization interface, followed by a short-linker and a β-hairpin strand tilted up ~45° from the membrane plane (Fig. 2a). We were able to refine the overall position of the short linker and β-hairpin strand predicted from the AlphaFold2[28] model (Fig. 2a). The loop domain reconnects to the beginning of TM3 via a short extracellular helix (ECH1), which was well supported by the cryo-EM maps (Fig. 2a). The β-hairpins were predicted by AlphaFold2[28] to be domain-swapped, and this structure refines well into the map density. The centre of the loop domain has a cluster of positively-charged residues K105, K107 and R108 from each of the two protomers (Fig. 2a). Overall, the TM2-TM3 β-loop domain clasps the two protomers together on the luminal side and forms a highly positively-charged cluster that is positioned above the dimerization domain interface. As such, the improved NHE9* structure is consistent with its previously proposed requirement for binding negatively-charged PIP₂ lipids at the interface[24].

## Cryo-EM structure of cysteine variant (NHE9*CC) shows binding of PI(3,5)P₂ lipids

To improve the resolution of the NHE9* structure we substituted L139 and I444 residues to cysteines in an attempt to disulphide-trap the inward-facing state. In addition, we added brain lipids during each step of the NHE9*CC affinity purification (see Methods). We subsequently collected a larger data-set of 13,780 movies and the final 3D-reconstruction had 78,370 particles from which an EM map was reconstructed to 3.2 Å according to the gold-standard Fourier shell correlation (FSC) 0.143 criterion (Supplementary Fig. 6a). Despite the similar FSC resolution estimate, these cryo-EM maps were improved with some local regions extending to 2.5 Å resolution (Supplementary Fig. 6a). In particular, the density was more defined for the TM2-TM3 β-loop at a lower map threshold (Supplementary Fig. 6a, b). Map density and sulphur atom distances, however, did not support disulphide bond formation between the two introduced cysteine residues (Supplementary Fig. 7a, b). As such, it is unclear whether the improved map quality seen using the NHE9*CC construct was because of the introduced cysteine residues and/or because of either the increased number of particles collected or the more frequent brain lipid addition during purification.

At the dimerization interface, we observed additional lipid-like features in the improved cryo-EM maps (Fig. 2b and Supplementary Fig. 7c). Rather than the cylindrical-like map density for fatty acids chains as seen in the NHE1 nanodisc structure with POPC lipids[27], the cryo-EM maps showed extended head-group lipid densities with distinct features (Fig. 2b, Supplementary Fig. 7c). Given that NHE9* co-purified from yeast with several lipids corresponding to the molecular mass of PIP₂[24], we attempted to model two PIP₂ lipids into the NHE9* CC maps and found that the density was a better fit for the phosphatidylinositol-3,5-bisphosphate PI(3,5)P₂, lipid, rather than the more

abundant phosphatidylinositol-4,5-bisphosphate PI(4,5)P₂ lipid (Fig. 2b, Supplementary Fig. 7c). Although PI(3,5)P₂ is a minor PIP₂ lipid, it is specific to endosomes, which overlaps with the functional localization of NHE9[35]. In contrast, the PI(4,5)P₂ lipid is principally located in the plasma membrane[36]. The glycerol backbone and acyl chains of the PI(3,5)P₂ lipid form hydrophobic stacking interactions to W321 (TM8, TM8') (Fig. 2c). Two glutamine residue Q17 (TM1, TM1') and N136 (TM3, TM3') are well positioned to hydrogen-bond with the C3-phosphate group (Fig. 2c). The position of these polar residues could partially explain the preference for PI(3,5)P₂ lipids, since there are no polar residues close enough to interact with a phosphate group at the C4-OH position.

Masked refinement of NHE9*CC was required to obtain high resolution maps to model PI(3,5)P₂ lipids, but map features for the TM2-TM3 β-hairpin loop domain could not be retained, likely as this region is still too dynamic (Supplementary Fig. 6a). Nevertheless, at lower contour levels, the map density for the β-hairpin loop domain matches the map density seen in the NHE9* maps at pH 6.5 (Supplementary Fig. 6a). Following a rigid-body fit of the TM2-TM3 β-hairpin loop domain in the NHE9*CC maps, the positively-charged loop domain residues K105, K105' K107, K107' were aligned to help neutralize the negatively-charge headgroups of PI(3,5)P₂ (Fig. 2d). Consistently, mutation of these lysine residues to glutamine was previously shown to abolish PIP₂-induced thermostabilization of NHE9[24].

## Assessing the requirement of PI(3,5)P₂ lipid binding to NHE9

We had previously probed interactions with PIP₂ and PIP₃ lipids to NHE9* using FSEC-TS and GFP-based thermal stability assays[18,24,37], an approach we had earlier confirmed capable of detecting cardiolipin-specific stabilization of *E. coli* NhaA[18,38]. The average melting temperature ($\Delta T_m$) of NHE9* increased by 8 °C following PI(4,5)P₂ addition, whereas other lipids POPC, POPE and POPA showed no clear thermostabilization[24]. To validate that PI(3,5)P₂ would also stabilise NHE9*, GFP-TS melting curves were produced using either PI(4,5)P₂ or PI(3,5)P₂ lipids (Methods and Fig. 3a). Consistently, NHE9* displayed higher thermostabilization after PI(3,5)P₂ addition with a $\Delta T_m$ of 15 °C, as compared to $\Delta T_m$ of 8 °C for PI(4,5)P₂ addition. Without lipid addition, NHE9* unfolds around 30 °C with a shallow slope for the transition temperature, which is indicative of a mixed-protein population in detergent (Fig. 3a). In contrast, with PI(3,5)P₂ added, NHE9* unfolded with a sharper transition, indicating it had shifted to a more homogenous protein population (Fig. 3a).

In the presence of various lipids, we further assessed the relationship between resistance to heat denaturation and retention of the NHE9* homodimer by FSEC[37] (Methods). As anticipated, we observed that a higher fraction of the NHE9* homodimer was retained if PI(3,5)P₂ lipid was added prior to heating at 50 °C for 10 min, as compared to addition of either PI(4,5)P₂ or POPC lipids (Fig. 3b). We had previously shown that substitution of β-hairpin TM2-TM3 lysine residues to glutamine in NHE9* K85Q-K105Q-K107Q abolished PI(4,5)P₂ stabilization and oligomerization using native MS[24]. Consistently, upon detergent extraction in crude membrane the NHE9* K85Q-K105Q-K107Q variant shows both dimer and monomer species and when purified further only a monomer species is observed (Supplementary Fig. 8a, b). The addition of either PI(3,5)P₂, PI(4,5)P₂ or POPC lipids to the purified NHE9* K85Q-K105Q-K107Q variant showed no clear thermostabilization (Supplementary Fig. 8c). Taken together, thermostability-shift assays confirms that NHE9* binds lipid PI(3,5)P₂ and its addition stabilises the functional homodimer in detergent.

NHE9 is localized to endosomes[5]. If PI(3,5)P2 lipids are required for NHE9 activity, then the protein might be inactive in membranes lacking this lipid. To assess the requirement of NHE9 activity for endosomal-specific lipids, we expressed human NHE9 and plasma membrane localized human NHE1 in a PS120 cell line, which is deficient in plasma membrane NHEs (Methods). In this mutant cell line, plasma

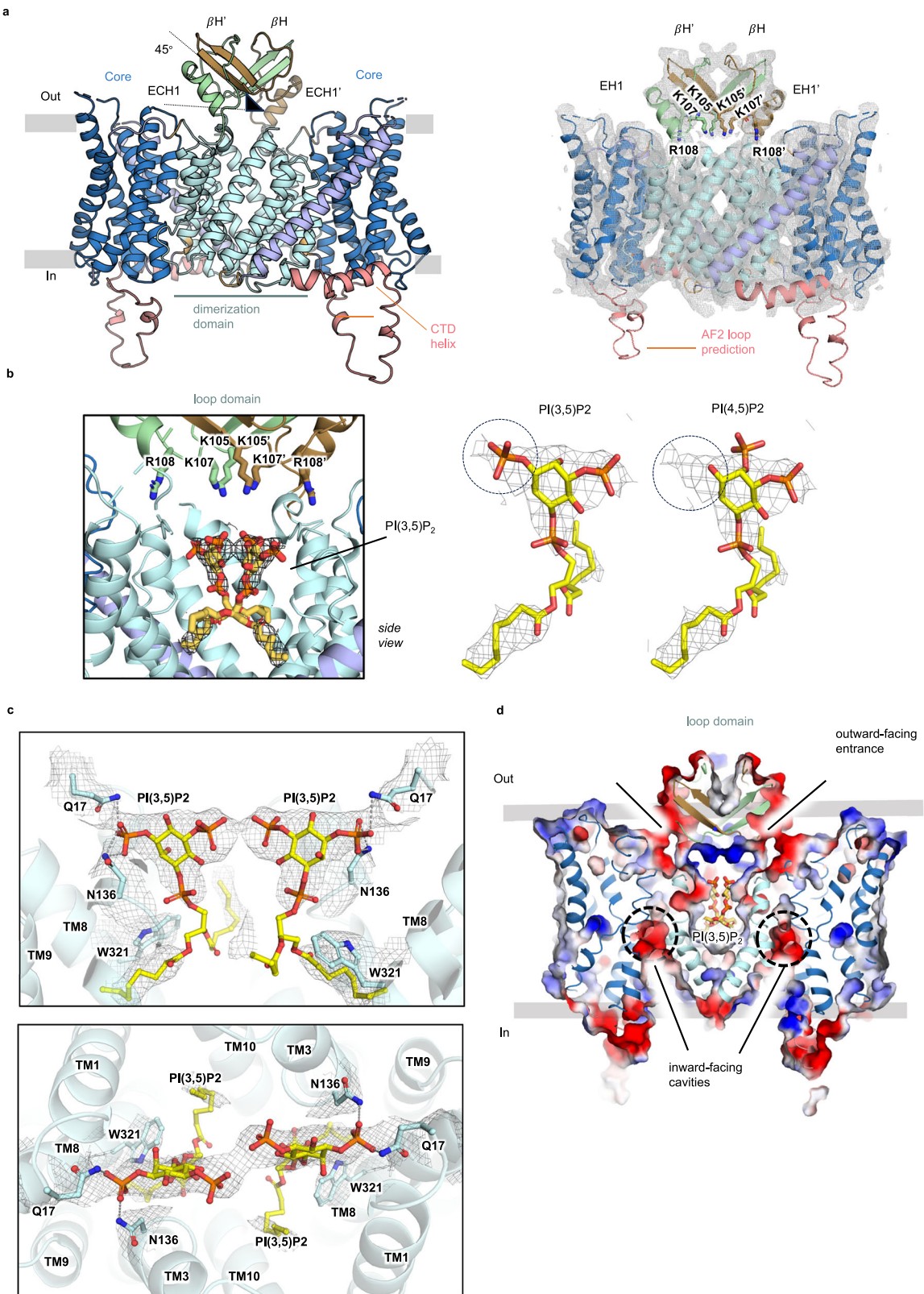

membrane localization of both human NHE1 and NHE9 isoform was confirmed after their transient transfection by surface biotinylation and streptavidin affinity isolation (Fig. 3c and Methods). To measure human NHE1 and NHE9 activity, the pH-sensitive dye BCECF was trapped intracellularly and pH recovery after acid loading with $NH_4Cl$ assessed spectroscopically. Whilst sodium-dependent intracellular pH recovery was clearly observed in the PS120 cells expressing human

NHE1, no NHE activity could be detected for human NHE9, despite similar expression levels (Fig. 3c, d). In contrast, human NHE9 activity was recorded during recycling endosome pH measurements of transfected PS120 cells (Fig. 3d, e). Thus, we can confirm that human NHE9 is inactive when mistargeted to the plasma membrane.

As far as we are aware, it is practically infeasible to incorporate $PI(3,5)P_2$ lipids into the plasma membrane of cultured cells and we

**Fig. 2 | TM2-TM3 β-loop domain structure of NHE9* CC with negatively-charged PI(3,5)P₂ lipids bound at the dimer interface. a** *left:* Structure of the NHE9*CC structure determined by cryo-EM and guided by the refined NHE9* model for the β-hairpin TM2-TM3 loop domain. Domain-swapped βH, βH′ and ECH1, ECH1′ in green and brown respectively, and NHE9 core and dimer domains. An additional C-terminal helix in the CTD could be modelled (salmon, cartoon), but there was no density to support the 20 residue AF2 loop model located between the end of the core domain and the beginning of the interfacial helix *right:* The cryo-EM maps superimposed on the NHE9* structure including the positively charged residues in the loop domain (sticks). **b** *left:* In the NHE9* CC structure additional map density supported the modelling of two PIP₂ lipids (yellow sticks) located in the middle at the dimerization interface between the two protomers and these could interact with the positively charged β-hairpin TM2-TM3 loop domain residues (green and brown sticks). *right:* The fit of PI(3,5)P₂ vs PI(4,5)P₂ lipids (sticks) into the cryo-EM maps is compared. **c** In addition to the β-hairpin TM2-TM3 loop domain lysine residues, PI(3,5)P2 is coordinated at the dimerization interface by the aromatic and polar residues Trp321, Gln17, Asn136 (sticks, with interactions illustrated by dashed grey lines). TM segments TM1 and TM8 are not shown for better visualization of the bound lipids (yellow sticks, map as grey mesh). **d** Electrostatic surface potential map (colored blue: positive to red: negative) cross-section with two negatively-charged PI(3,5)P₂ lipids modelled in the hydrophobic and positively-charged dimerization interface. The interface between the loop domains and core domains (indicated by the black-line) are negatively-charged and provide an electrostatic pathway for cations in the outward-facing state. Protein shown as cartoon (colored as in a), PI(3,5)P₂ as sticks (yellow) and the ion-binding site is highlighted (dotted-circle).

therefore required another approach to assess whether NHE9 activity is dependent on PIP₂ lipids. Previously, NHE9* was reconstituted into liposomes together with $F_0F_1$-ATP synthase for proteoliposome studies and an apparent $K_M$ of NHE9* for Na⁺ of 20.5 ± 3 mM was determined[24]. However, a high background from empty liposomes would make assessing lipid requirements for NHE9* challenging in this set-up, as the signal-to-noise ratio was previously low ~3:1 and the bacterial ATPase protein used has different lipid preferences to NHE9. More recently, we have used solid-supported membrane (SSM) electrophysiology to record Na⁺ translocation of SLC9B2 and SLC9C1 Na⁺/H⁺ exchangers[15,16]. In this setup, proteoliposomes containing NHEs are adsorbed to a SSM and charge translocation is measured via capacitive coupling to the supporting membrane bilayer[39].

We forthwith recorded SSM responses for NHE9* proteoliposomes upon the addition of 20 mM NaCl (Fig. 4a and Methods). Peak currents were ~6-fold higher than either the NHE9* variant for which ion-binding site residues N243 and D244 were substituted to alanine, or the unrelated mammalian transporter for fructose (GLUT5) (Fig. 4a). The apparent binding affinity ($K_d$ ᵃᵖᵖ) determined by SSM for Na⁺ for NHE9* was 36.3 ± 4 mM and NHE9*CC was 24.1 ± 8 mM, which were similar to the Michaelis−Menten $K_M$ estimate for NHE9* in proteoliposomes (Fig. 4b, c)[24]. Thus, we could confirm that SSM can be used to monitor Na⁺ binding to NHE9*, and that the protein binds sodium at a similar concentration range to the in vivo $K_M$ estimates measured for the intracellular localized NHE6 and NHE8 isoforms[1].

To assess the influence of PI(3,5)P₂ lipid to NHE9* activity, we incubated the purified NHE9* protein with either buffer or buffer containing solubilised PI(3,5)P₂ lipids, and reconstituted the respective mixtures into liposomes made from yeast-polar lipids (Methods). SSM-based electrophysiology of NHE9* incubated with buffer produced similar peak currents upon Na⁺ addition to those shown previously, whereas NHE9* protein incubated with PI(3,5)P₂ lipid produced a stronger response to Na⁺ (Fig. 4d). Consistently, we observed a four-fold increase in its apparent binding affinity for Na⁺ ($K_d$ ᵃᵖᵖ = 9.3 mM) (Fig. 4d–e). In contrast, PI(4,5)P2 lipid addition had a similar Na⁺ binding affinity as the buffer only samples (Fig. 4e). To confirm that the response to PI(3,5)P₂ addition was mediated by the expected interaction with the lysine residues in the TM2-TM3 β-hairpin loop domain, we recorded SSM-based currents for the purified lysine-to-glutamine NHE9* K85Q-K105Q-K107Q variant. The triple glutamine variant also showed a weaker apparent affinity for Na⁺ ($K_d$ ᵃᵖᵖ = 55 mM) compared with NHE9* and, in fact, the addition of the PI(3,5)P₂ lipid weakened its apparent affinity for Na⁺ ($K_d$ ᵃᵖᵖ = 88 mM) (Fig. 4f). The outside surface of the PI(3,5)P₂-stabilised NHE9* TM2-TM3 β-hairpin loop domain has a partially negatively-charged surface (Fig. 2d). Is seems that stabilization of the TM2-TM3 β-hairpin loop domain by PI(3,5)P₂ lipids provides an electrostatic pathway that can better attract Na⁺ to the outward-facing funnel.

Despite the PI(4,5)P2 lipid inducing thermostabilization of NHE9* in detergent, no increase in the affinity for Na⁺ was observed. It is plausible that the affinity for PI(4,5)P2 lipid is weaker than PI(3,5)P2 lipid and, so when incorporated into liposomes, other lipids can effectively compete for PI(4,5)P2 binding. Alternatively, PI(4,5)P2 may be able to partially stabilise the NHE9 dimer, but does not properly engage with the β-hairpin loop TM2-TM3 domain. To further evaluate how the TM2-TM3 β-hairpin loop domain could interacts with modelled PIP₂ lipids, we carried out molecular dynamics (MD) simulations of the NHE9*CC structure and a in silico model for an NHE9* K85Q-K105Q-K107Q variant (see Methods).

Using 500 ns all-atoms MD simulations, the PI(3,5)P₂ lipid remains within 3 Å from its previous 10 ns position for nearly the entire simulation time, suggesting the stability of the lipid headgroup within the binding pocket (Fig. 5a, b). In contrast, a modelled PI(4,5)P₂ lipid was less stably bound with a higher fraction of frames showing >3 Å movement from its previous 10 ns frames, suggesting a less stable interaction (Fig. 5b and Supplementary Fig. 9a). The distribution of salt-bridge interactions with K105 and K107 residues revealed that K105 residues were nearly always in contact with the PI(3,5)P₂ lipid, whereas K107 residues remained in contact for around half of the simulation time (Fig. 5c–e). The K105 and K107 are poised to form salt bridges with the phosphates and, hence, neutralise the charge within the PI(3,5)P₂, allowing two PI(3,5)P₂ to be in close proximity to each other (Supplementary Fig. 9b). The PI(4,5)P₂ lipid had a much larger variation for interaction with the lysine residues and the preferred interaction with K105 was less pronounced (Fig. 5d, e). In MD simulations of the modelled NHE9* K85Q-K105Q-K107Q variant, neither of the PIP₂ lipids were stably bound and the lipids had no clear interaction with the modelled glutamines (Fig. 5b–d). Thus, the polar and aromatic residues located at the dimerization interface that, together with positive-charged residues from the TM2-TM3 β-hairpin loop, create an environment better suited for binding the endosomal specific PI(3,5)P₂ lipid.

## Further mechanistic insights from the improved cryo-EM NHE9 CC* structure

In the improved NHE9*CC maps, we were able to model all side-chains forming the ion-binding site, which are positioned around the half-helical cross-over (Fig. 6a–c). We observed some minor differences between the side-chain positions of T214, D215, E239 and R408 from the previously reported NHE9* structure[24], and also between NHE9 and NHE1 structures (Fig. 6b–c). In MD simulations of NHE9, Na⁺ could be coordinated within the core domain forming interactions to D244, N243, S240 and T214 and several waters[24]. Based on phylogenetic analysis[40], it was predicted that a salt-bridge would also form between E239 in TM6 and R408 in TM11. Here, we can confirm a salt-bridge is indeed formed between E239 and R408 residues and given the position of E239 in the ion-binding site, we propose this salt-bridge aids the stabilization of the residues required for coordinating Na⁺ (Fig. 6c). Moreover, T214 forms a parallel hydrogen bond to N243 (Fig. 6c). This unexpected hydrogen bond establishes a more rigid ion-binding site

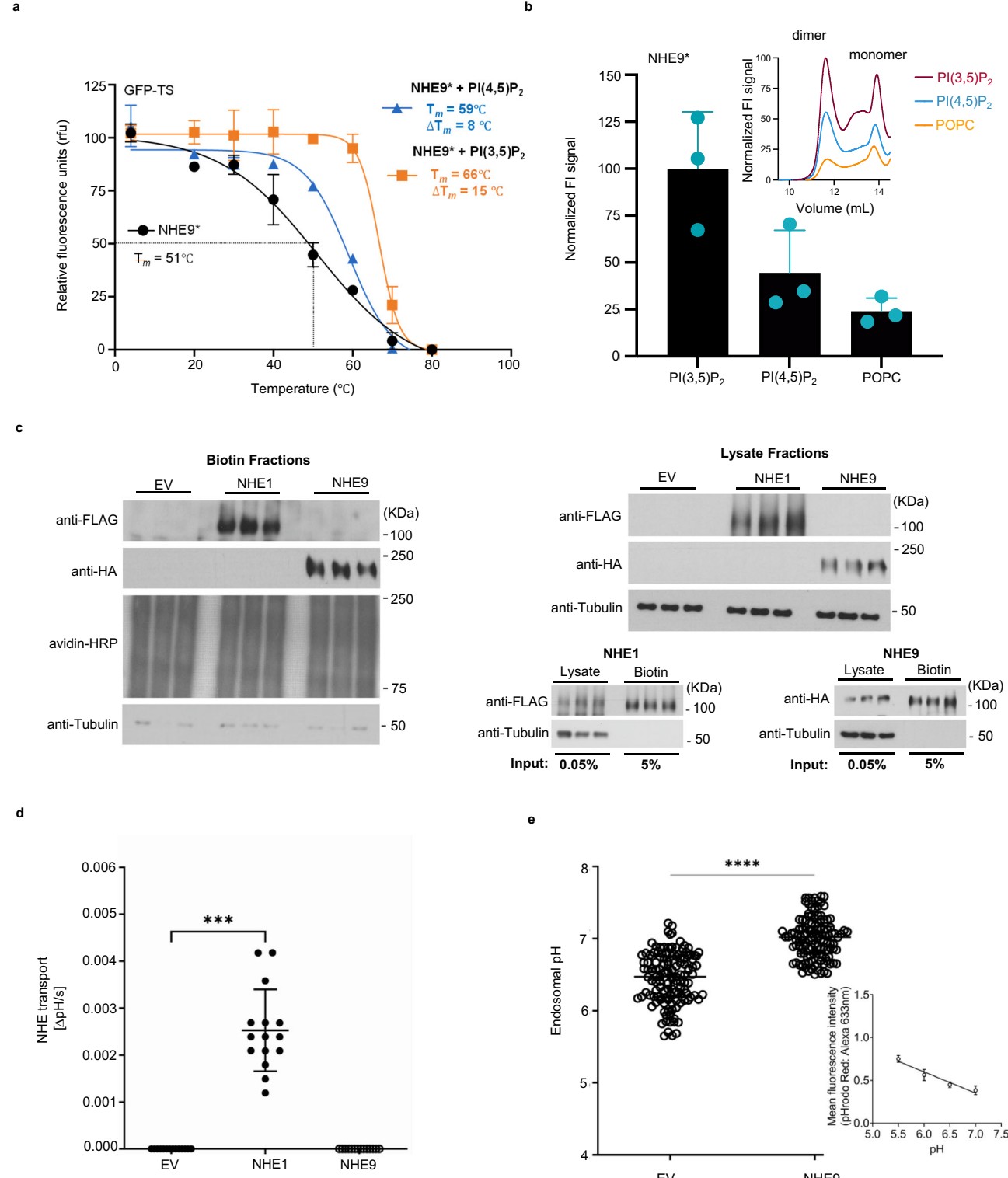

than that observed in NHE1, since the T214 residue is replaced by valine in NHE1 (Fig. 6b–c). Indeed, all the plasma membrane localized NHE isoforms have a hydrophobic residue in this position, whereas the intracellular isoforms have a threonine residue[24]. It is plausible that these ion-binding site differences will help to enable intracellular isoforms to also transport K[+141,] but this will need more thorough investigation.

In NHE proteins, extrinsic factors bind to a large, intracellular C-terminal tail of ~125–440 amino acids in length[4,7,42,43]. In NHE1,

removal of the C-terminal tail results in a constitutively active transporter[44]. The C-terminal tail has been referred to as an allosteric regulatory subunit, which influences ion-exchange activity upon interaction with many effectors[45] e.g., calmodulin (CaM)[41,43,46–48]. The recent human NHE1 structure in complex with Calcineurin B homologous protein 1 (CHP1) revealed an interaction with an interfacial α-helical stretch formed by residues 517 to 539[27]. However, CHP1 interacting with the interfacial helix was found to have no direct contacts with the transporter module itself, and it is currently unclear how CHP1

**Fig. 3 | NHE9 is non-functional at the plasma membrane and the endosomal-specific lipid PI(3,5)P$_2$ stabilizes the functional homodimer and activity.**
**a** Thermal shift stabilization of purified dimeric NHE9*GFP in the presence of PI(4,5)P$_2$ (blue) and PI(3,5)P$_2$ (mustard) compared to lipid-free (black). Data presented are normalized fluorescence of mean values ± s.d of n = 3 independent experiments; the apparent $T_M$ was calculated from datapoints fitted according to a sigmoidal 4-parameter logistic regression function. **b** Lipid stabilization of NHE9*. Normalised fraction of NHE9* homodimer remaining after heating at 50 °C for 10 min in the presence of DDM-solubilised lipids and are s.e.m. of $n$ = 3 independent experiments, *top inset*: representative FSEC traces of the tabulated data shown. **c** *left*: Plasma membrane expression of FLAG-tagged NHE1 and HA-tagged NHE9 in PS120 cells. PS120 cells were transfected with empty vector (EV; pMH), or vector encoding human NHE1 or human NHE9. Plasma membrane proteins were isolated using surface biotinylation. Avidin−HRP was used as loading control, *right above*: Immunoblots of PS120 cell lysates depicted in the left panel, *right below*: Immunoblots of PS120 cell lysates and biotin fractions for FLAG-tagged NHE1 (left panel)

and HA-tagged NHE9 (right panel). Indicated is the percentage of the input loaded on the SDS-page gel. Three independent biological replicates were used in each of the experiment per condition. **d** Measurement of plasma membrane NHE1 transport activity with empty vector (EV; negative control), human NHE1 (positive control), or human NHE9-transfected PS120 cells was performed by quantifying the sodium-dependent intracellular pH recovery after cytoplasmic acidification represented as mean values ± SD pooled from 3 independent experiments with 5 biological replicates each. Statistical analysis was performed using the Kruskal-Wallis test with Dunn's post-hoc test for multiple comparisons. EV vs. NHE1 ($P$ = 0.0002) and EV vs. NHE9 ($P$ = 0.2968). e Recycling endosome pH measurement of PS120 cells transfected with empty vector (EV; negative control) or human NHE9. Each dot represents measurement of an individual cell ($N$ = 131 observations for EV transfected cells and $N$ = 126 observations for NHE9 transfected cells). Asterisk denotes significance for the indicated comparison (two-tailed unpaired Student's *t*-test; ****$p$ < 0.0001). *inset*: pH calibration curve for endosomal pH measurements.

binding increases NHE1 activity[27]. Key to developing an allosteric model for extrinsic regulation is to obtain an NHE structure without complex partners. In the improved NHE9*CC structure, we could model part of the C-terminal tail without protein complex partners (Fig. 6d, Supplementary Fig. 11). The interfacial helix in the C-terminal domain (CTD) sits on the membrane interface and wraps around from the core domain to the linker helix TM7 (Fig. 6d). In particular, the highly-conserved K301 on the linker helix makes direct hydrogen bond interactions to carbonyl of L540, backbone amine of T541 and carbonyl of T541 (Fig. 6d, e). Given that the NHE1 and NHE9 structures superimpose well, apart from the position of the interfacial helix, it seems likely that the binding of CHP1 has driven its dissociation from the linker helix (Fig. 6f, g). Consistently, without CHP1 present, the AlphaFold2[28] model of NHE1 is similar to NHE9, with clear interactions observed between TM7 and the CTD helix (Fig. 6h). The cytoplasmic surface is also more positively charged with the CTD helix in the likely inhibited position, which could diminish the attraction of positively-charged ions to the inward-facing cavity (Supplementary Fig. 11a−c).

## Discussion

Elevator transporters are distinguished from rocker-switch and rocking bundle transporters by the fact that substrates are translocated by just one of the two domains[10]. The transport domain is able to move independently from the scaffold domain and carry the substrate across the membrane, as the scaffold domain does not participate in substrate binding. It seems in elevator proteins that oligomerization is likely required for the transporter domain to move effectively against the scaffold domain. In some elevator proteins, such as Na$^+$-coupled glutamate transporters, the scaffold domain is extensive and, once formed, oligomerization is thought to be lipid insensitive[49]. In the Na$^+$/H$^+$ exchanger family, however, the scaffold domain can be flexible and shows large structural variations[24], thus implying a role susceptible to regulation. Indeed, in the Na$^+$/H$^+$ exchanger NHA2 (SLC9B2) the additional N-terminal helix can readjust its position in the presence of yeast PI-lipids to make a more compact homodimer[15]. Moreover, monomeric mutants of NHA2 are inactive and thus indicated that lipid-dependent stabilization of the homodimer is likely required for functional activity[15], although this is yet to be firmly established.

Here, we have compared the lipid-mediated oligomerization in the Na$^+$/H$^+$ exchanger NhaA and the endosomal exchanger NHE9, which both harbour β-hairpins between topologically equivalent helices in the scaffold domain. The scaffold β-hairpin loop domain is absent is the bacterial Na$^+$/H$^+$ members NapA[12] and NhaP[13], and in mammalian NHE isoforms that are localised to the plasma membrane[27,50]. Moreover, the high structural-similarity between plasma membrane localised NHE1 and endosomal NHE9, implies that the additional TM2-TM3 β-hairpin loop domain likely possess a regulatory role. Here we could structurally confirm that CDL binds

between the NhaA protomers and it is there stabilised by interactions with arginine residues. In addition, CDL coordination further involves a tryptophan residue located at the end of one of the scaffold helices. Consistently, in computational-based screening for CDL binding to bacterial proteins, a tryptophan residue seems to be a key requirement for ligand-like lipid interactions[32]. In NHE9, we observe the PI(3,5)P2 lipid interacting also with a tryptophan residue from one of the scaffolding helices at the dimerization interface. The coordination of PI(3,5)P2 is further established by polar residues at the dimer interface and the positively-charged residues in the TM2-TM3 β-hairpin loop. Although the coordination for binding CDL and PI(3,5)P2 is clearly different, in both structures the respective lipids interact with residues of the topologically-equivalent scaffold helices, namely TM2 in NhaA and TM3 in NHE9. This scaffold helix harbours highly-conserved hydrophobic gating residues[24] that are located opposite to the ion-binding site residues in the core domain. Their role seems to be to prevent elevator transitions, unless the negatively-charged aspartate has bound its substrate ion[24]. Thus, we propose a common coupling between the ion-binding, and the lipid-stabilized oligomerization, whereby stabilization of the scaffold helix enhances Na$^+$ binding by making a more favourable pathway − indeed, the outside surface of the loop domain in NHE9 is also negatively charged (Fig. 2d). As yet, it is unclear whether catalytic turnover for Na$^+$ would also increase with an enhanced stability of the scaffold domain.

The structure of NhaA in complex with CDL supports the proposed regulatory switch induced by salt-stress in *E. coli*[18]. In NHE9, the specific binding of PI(3,5)P$_2$ is consistent with its subcellular location, and the lipid could either enhance or completely turn on NHE9 activity by stabilizing the homodimer once the protein reaches endosomes (Fig. 7). Consistent with a regulatory switch, PI(3,5)P$_2$ is a minor lipid that is not found in the plasma membrane, but only found in endosomes and lysosomes[35,36]. Moreover, the V-type ATPase is known to co-localise with NHE9 in endosomes, and work together to fine-tune organellular pH[25]. Consistent with the proposed model, the activity of the V-type ATPase is also increased in the presence of PI(3,5)P$_2$ lipids[51]. Lastly, inhibition of enzymes required to produce PI(3,5)P$_2$ lipids results in impaired epidermal growth factor receptor (EGFR) trafficking[52]. Indeed, NHE9 activity has been shown to be critical for EGFR sorting and turnover[25]. Since intracellular NHEs recycle through the plasma membrane they could, in principle, acidify the cell upon exposure to the high Na$^+$ levels outside the cell, and yet vesicular acidification in NHE7-expressing cells has not been measured[53] and herein NHE9 was likewise found to be inactive at the plasma membrane. It is likely that hydrophobic mismatch will drive dimer formation of the shorter scaffold helices of NHE9 in the ER[54]. Our working model is that dimer formation in NHE9 is retained upon trafficking, yet the scaffold domains are too flexible without PI(3,5)P2 binding, to actually catalyse robust ion-exchange. To the best of our knowledge, such a

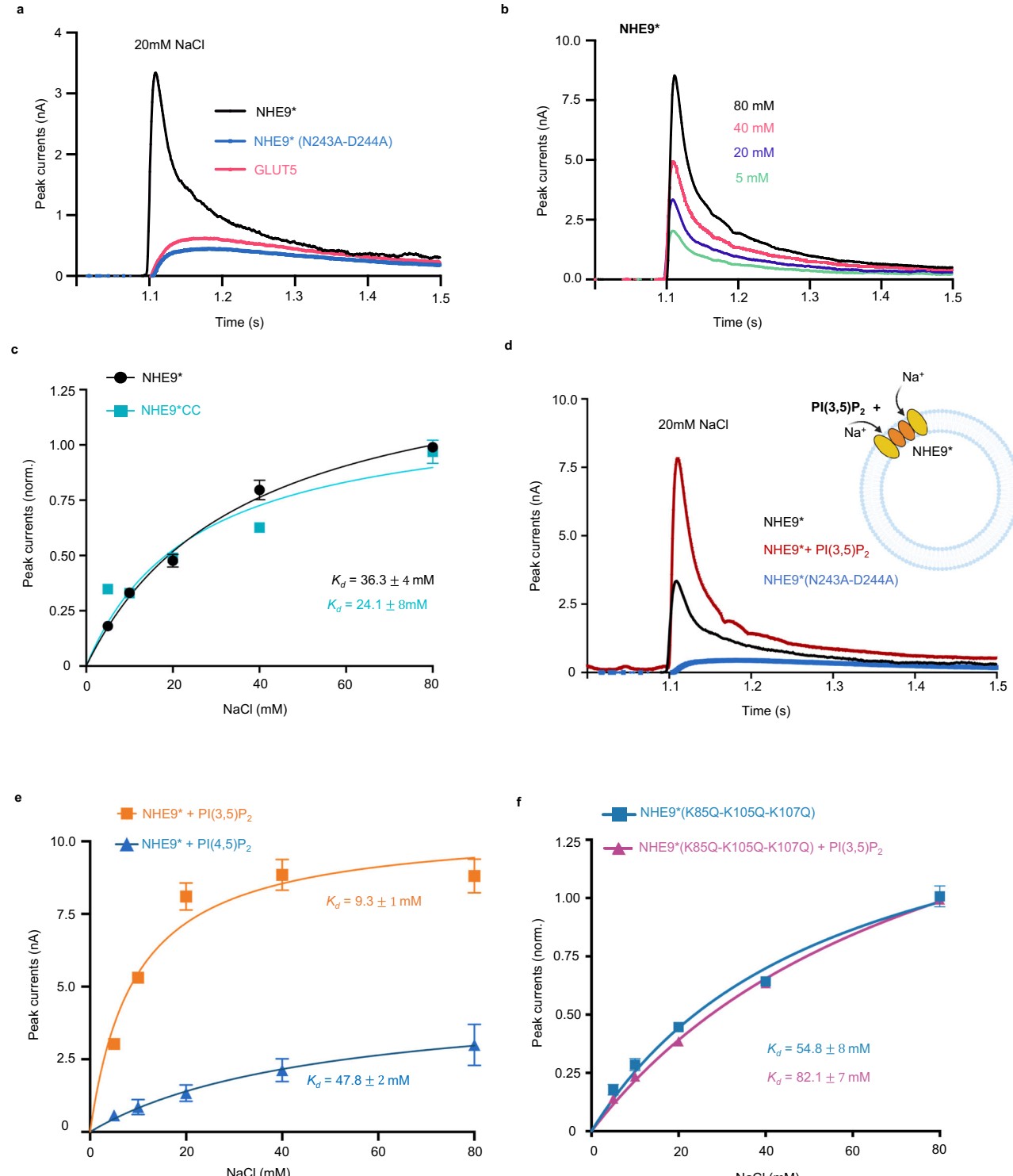

**Fig. 4 | SSM-based electrophysiology measurements of NHE9\* and variants in proteoliposomes. a** Transient currents were recorded for NHE9\* proteoliposomes (black trace) at symmetrical pH 7.5 after the addition of 20 mM NaCl. Peak currents for an ion-binding site NHE9\* variant N243A-D244A (blue trace) and fructose transporter GLUT5[80] (pink trace) after the addition of 20 mM NaCl are also shown. **b** SSM-based electrophysiology measurements of NHE9\* proteoliposomes with transient currents recorded after Na⁺ concentration jumps at pH 7.5 on both sides. **c** Fit of the normalized amplitude of the transient currents for NHE9\* and NHE9\*CC as a function of Na⁺ concentrations and the corresponding binding affinity across $n = 3$ different sensors (K_d), respectively. Error bars are the mean ± s.d. for $n = 3$ independent experiments (sensors). **d** Transient currents were recorded for NHE9\* proteoliposomes (black trace) at symmetrical pH 7.5 after the addition of 20 mM NaCl. Peak currents for an ion-binding site NHE9\* variant N243A-D244A (blue

trace) and NHE9\* pre-incubated with synthetic PI(3,5)P₂ lipids (red trace) are also shown. *above:* schematic of NHE9\* and PI(3,5)P₂ reconstituted into liposomes for SSM based electrophysiology measurements, figure created using Biorender.com **e.** Fit of the amplitude of the transient currents as a function of Na⁺ concentrations at pH 7.5 for NHE9\* proteoliposomes pre-incubated with synthetic PI(3,5)P₂ lipids (orange trace) and synthetic PI(4,5)P₂ lipids (blue trace) with their corresponding binding affinity ($K_D$). Error bars are the mean ± s.d. for $n = 3$ independent experiments (sensors). **f** Fit of the normalized amplitude of the transient currents for NHE9\* β-hairpin lysine variant (K85Q, K105Q, K107Q) pre-incubated with buffer (blue curve) or PI(3,5)P2 lipid (purple) as a function of Na⁺ concentrations and the corresponding binding affinity across 3 different sensors (K_d). Error bars are the mean ± s.d. for $n = 3$ independent experiments (sensors). Created in BioRender. Kokane, S. (2025) https://BioRender.com/m80i164.

lipid-conditioned-activation model would be a previously unobserved regulatory mechanism for ion-transporters and SLCs in general.

In addition to the lipid-dependent oligomerization, mammalian NHEs have a long C-terminal tail that regulates their activity[1]. The C-terminal domain is poorly conserved across the different NHE members and, so far, only part of the C-terminal tail could be modelled for human NHE1 and NHE3 in complex with CHP1[27,50]. Structures of human NHE1-CHP1 in outward and inward-facing states show some differences in the position of the C-terminal tail[27]. Based on these structural differences, it was proposed that CHP1 may increase activity by favouring an outward-facing state. In the improved NHE9*CC structure we could model the corresponding region of the C-terminal tail, but in the absence of any interacting regulatory proteins. Surprisingly, we find that NHE9 has a similar C-terminal interfacial helix to that seen in NHE1 and NHE3[27,50]. However, in the absence of a regulatory protein, the C-terminal interfacial helix in NHE9*CC is making a number of direct contacts with the linker helix TM7 through interaction with a lysine residue, likely restricting its mobility and that of the core transport domains. An autoinhibitory role of the C-terminal tail has been previously proposed for NHE1[55], this effect was removed upon $Ca^{2+}$-calmodulin binding to a site distal to the CTD helix[56]. Comparing our structure of NHE9 with NHE1-CHP1 provides a mechanistic framework for the autoinhibitory model, in that the repositioning of an auto-inhibitory C-terminal tail by binding partners would remove the inhibition and enable ion-exchange. Nevertheless, there is likely to be further complexity to this conceptual model, since positive regulators like $Ca^{2+}$-calmodulin, can both displace this C-terminal helix and promote homodimerization by itself forming oligomers[56]. Furthermore, $PIP_2$ lipids can also with interact the C-terminal tail[57] and, together with phosphorylation[44], this will likely influence the mobility of the CTD and thus the degree of autoinhibition. Indeed, the complexity for auto-regulation was more recently revealed with the structure of human NHE3, which was still in an auto-inhibited state with CHP1 bound as a distal loop, downstream of the CTD helix, had protruded into the inward-facing cavity[50].

In summary, our work provides the structural and molecular framework for the allosteric regulation of β-hairpin containing $Na^+/H^+$ exchangers by the modulation of the dimerization propensity from specific lipids binding at their interfaces. It was recently shown that the plasma membrane-localised elevator anion exchanger AE1 binds PI(4,5)P2 lipids between the scaffold domains of the homodimer[58,59], yet the $PIP_2$ lipid does in this case appear to enhance activity by displacing the N-terminal cytosolic domain[58]. Although the lipid-dependent regulation and C-terminal tail interactions are fine-tuned differently between transporter families, it is reasonable to assume that we will nonetheless see a convergence in similar models for their allosteric regulation. Thus, the regulatory themes revealed are likely to be also relevant to other small molecule transporters in general.

## Methods

### Expression and purification of EcNhaA-mut2
E. coli NhaA WT-like triple mutant (A109T, Q277G, L296M), with a TEV-cleavable C-terminal GFP-His$_8$ tag was overexpressed in the E. coli strain Lemo21 (DE3) and purified[22]. Briefly, the NhaA triple mutant was extracted from membranes with n-Dodecyl β-D-maltoside (DDM; Glycon) and purified by IMAC (Ni-NTA; Qiagen). To purified NhaA-triple-mutant-GFP fusion, a final concentration of 3 mM cardiolipin (18:1) in 0.05% DDM was added, and then dialyzed overnight against buffer consisting of 20 mM Tris-HCl pH 7.5, 150 mM NaCl and 0.03% DDM. The dialysed NhaA-triple-mutant-GFP fusion was subjected to size-exclusion chromatography and the peak concentrated to 3.5 mg.ml$^{-1}$.

### Expression and purification of NHE9*
The NHE9* structural construct was identified previously in ref. 24, and is partially truncated on the C-terminal tail consisting of residues 8 to

575 out of a total of 644. The constructs NHE9*CC the L139C-I444C double mutant, NHE9*(N243A-D244A), NHE9*(K85Q-K105Q-K107Q) were synthesized and cloned into the GAL1 inducible TEV-site containing GFP-TwinStrep-His$_8$ vector pDDGFP$_3$. The cloned NHE9* and its variants were transformed into the S. cerevisiae strain FGY217 and cultivated in 24-L cultures of minus URA media with 0.2% of glucose at 30 °C at 150 RPM Tuner shaker flasks using Innova 44 R incubators (New Brunswick). Upon reaching an $OD_{600}$ of 0.6 AU galactose was added to a final concentration of 2% (w/v) to induce protein over-expression. Following incubation at the same conditions the cells were harvested 22 h after induction by centrifugation (5000 × g, 4 °C, 10 min). The cells were resuspended in cell resuspension buffer (CRB, 50 mM Tris-HCl pH 7.6, 1 mM EDTA, 0.6 M sorbitol) and subsequently lysed by mechanical disruption[60]. Centrifugation (10,000 × g, 4 °C, 10 min) was used to remove cell debris. Membranes were subsequently isolated from the supernatant by ultracentrifugation (195,000 × g, 4 °C, 2 h), resuspended and homogenized in membrane resuspension buffer (MRB 20 mM Tris-HCl pH 7.5, 0.3 M sucrose, 0.1 mM $CaCl_2$).

For structural studies of NHE9* at pH 6.5, the membranes were extracted and purified as described previously[24]. In short, the Streptag-purified protein after removal of the C-terminal affinity-GFP-tag was collected and concentrated using 100 kDa MW cut-off spin concentrator (Amicon Merck-Millipore) further purified by size-exclusion chromatography (SEC), using a Superose 6 increase 10/300 column (GE Healthcare) and an Agilent LC-1220 system in 20 mM Mes-Tris pH 6.5, 150 mM NaCl, 0.003% (w/v) LMNG, 0.0006% (w/v) CHS. For NHE9*CC, the isolated membranes were solubilized as mentioned before[24] with addition of brain extract from bovine brain type VII (Sigma-Aldrich, cat. nr. B3635) to a total concentration of 0.003 mg/ml, in the solublization, wash and elution buffers[24]. The cleaved protein was collected and concentrated using 100 kDa MW cut-off spin concentrators (Amicon Merck-Millipore) separated by size-exclusion chromatography (SEC), using a Superose 6 increase 10/300 column (GE Healthcare) and an Agilent LC-1220 system in 20 mM Tris-HCl pH 8, 150 mM NaCl, 0.003% (w/v) LMNG, 0.0006% (w/v) CHS.

### Cryo-EM sample preparation and data acquisition and processing
3 μg of purified NHE9*CC sample was applied to freshly glow-discharged Quantifoil R2/1 Cu300 mesh grids (Electron Microscopy Sciences) and blotted for 3.0 s with a 20 s waiting time prior, under 100% humidity and subsequently plunge frozen in liquid ethane using a Vitrobot Mark IV (Thermo Fisher Scientific). Cryo-EM datasets were collected on a Titan Krios G3i electron microscope operated at 300 kV equipped with a GIF (Gatan) and a K3 Bioquantum direct electron detector (Gatan) in counting mode. The movie stacks were collected at 130,000× magnification corresponding to a pixel size of 0.66 Å in a counted super-resolution mode. All movies were recorded with a defocus range of −0.4 to −2.5 μm. Similarly, 2.4 μg of purified NHE9* at pH 6.5 was blotted and the movie stacks were collected at 165,000x magnification, with a pixel size of 0.82 Å on Titan Krios G2 electron microscope operated at 300 kV equipped with a GIF (Gatan) and a K2 summit direct electron detector (Gatan) in counting mode The movies were recorded with a defocus range of −0.9 to −2.3 μm. 3 μl of EcNhaA-mut2 with concentration of 3.5 mg/ml was blotted and the movie stack was collected at 130,000x magnification, with a pixel size of 0.6645 Å. The movies were recorded with a defocus range of −0.6 to −2.0 μm. The statistics of all cryo-EM data acquisition are summarized in Supplementary Table 1.

### Image processing NHE9* at pH 6.5
The dose-fractioned movies were corrected by using MotionCorr2[61]. The dose-weighted micrographs were used for contrast-transfer-function estimation by CTFFIND-4.1.13[62]. The dose-weighted images

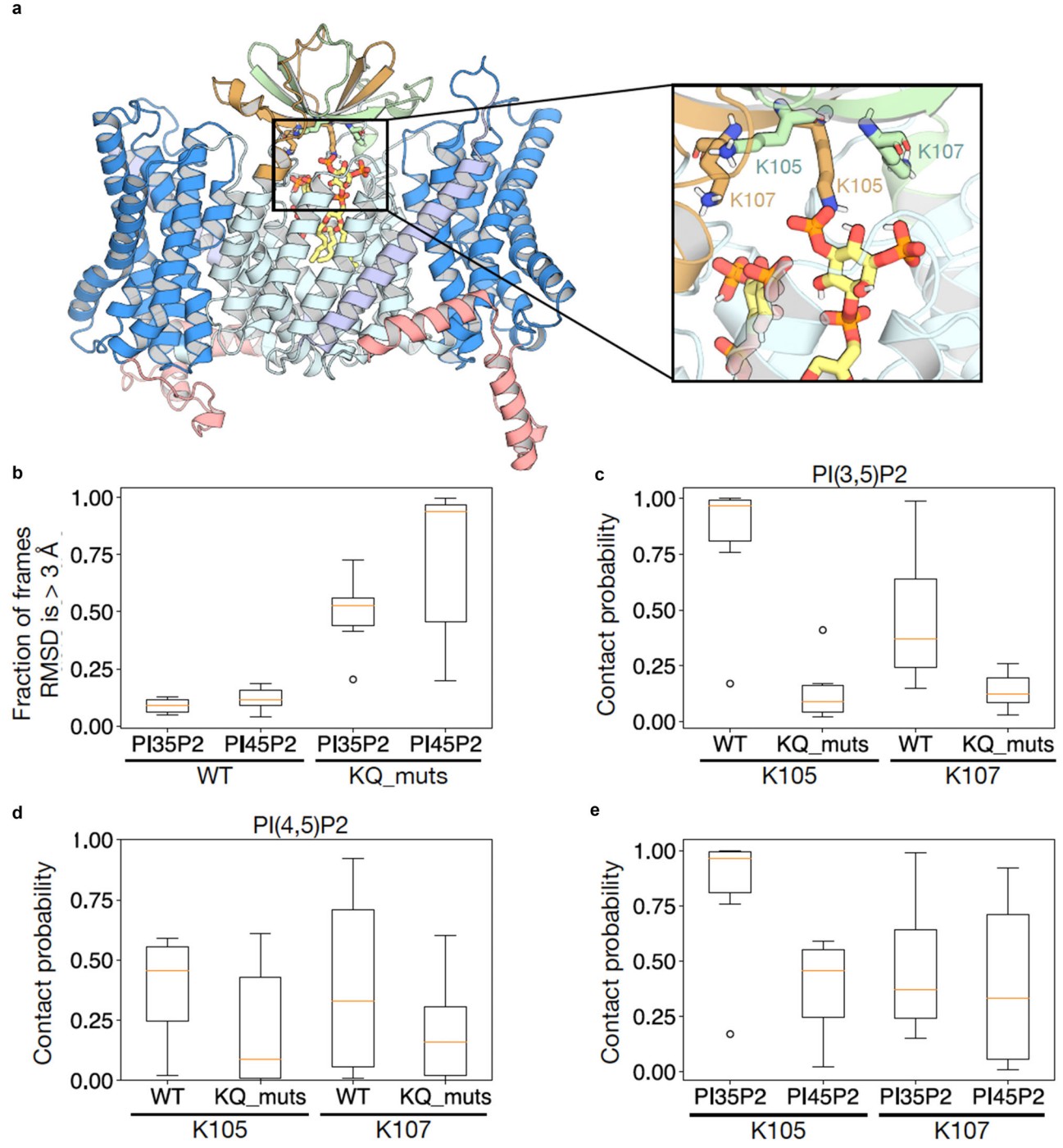

**Fig. 5 | MD simulations of PIP₂ interaction to NHE9\* CC. a** A representative snapshot of NHE9 with PI(3,5)P₂ after a 250 ns simulation. The structure of PI(3,5)P₂ is shown in yellow. K105 and K107 of the different protomers are shown in green and brown, respectively. Protein shown as cartoon and lipids as sticks (colored as in Supplementary Fig. 1d). **b** Box plot representation of the distribution of the fractions of frames where the root mean square deviation (RMSD) of PIP₂ lipids is > 3 Å with respect to its previous 10 ns structure. **c** Box plot representation of the distribution of frames where PI(3,5)P₂ are within 4 Å (defined as contact

probability) with either K105, K107 or its mutants. **d** Box plot representation of the distribution of frames where PI(4,5)P₂ are within 4 Å (defined as contact probability) with either K105, K107 or its mutants. **e** Box plot representation of the distribution of the contact probability between K105 and K107 with either PI(3,5)P₂ or PI(4,5)P₂. The box (b-e) plots are defined as follows: the center line represents the median, the bounds of the box correspond to the 25th and 75th percentiles, and the whiskers extend to the minimum and maximum values excluding outliers. Outliers are depicted as circles.

were used for auto-picking, classification and 3D reconstruction. Approximately 1000 particles were manually picked, followed by a round of 2D classification to generate templates for a subsequent round of auto-picking using RELION-3.0 beta[63]. The auto-picked particles were subjected to multiple rounds of 2D classification using RELION-3.0 beta to remove bad particles and "junk". The particles

belonging to "good" 2D classes were extracted and used for initial model generation using RELION-3.0 beta[63].

To visualize the extracellular loop domain of NHE9, the aligned 244,279 particles from RELION was imported into CryoSPARC[34]. *UCSF pyem*[64] was used for file format conversion from RELION to CryoS-PARC. 3D Variability Analysis (3DVA) in CryoSPARC[34] was performed

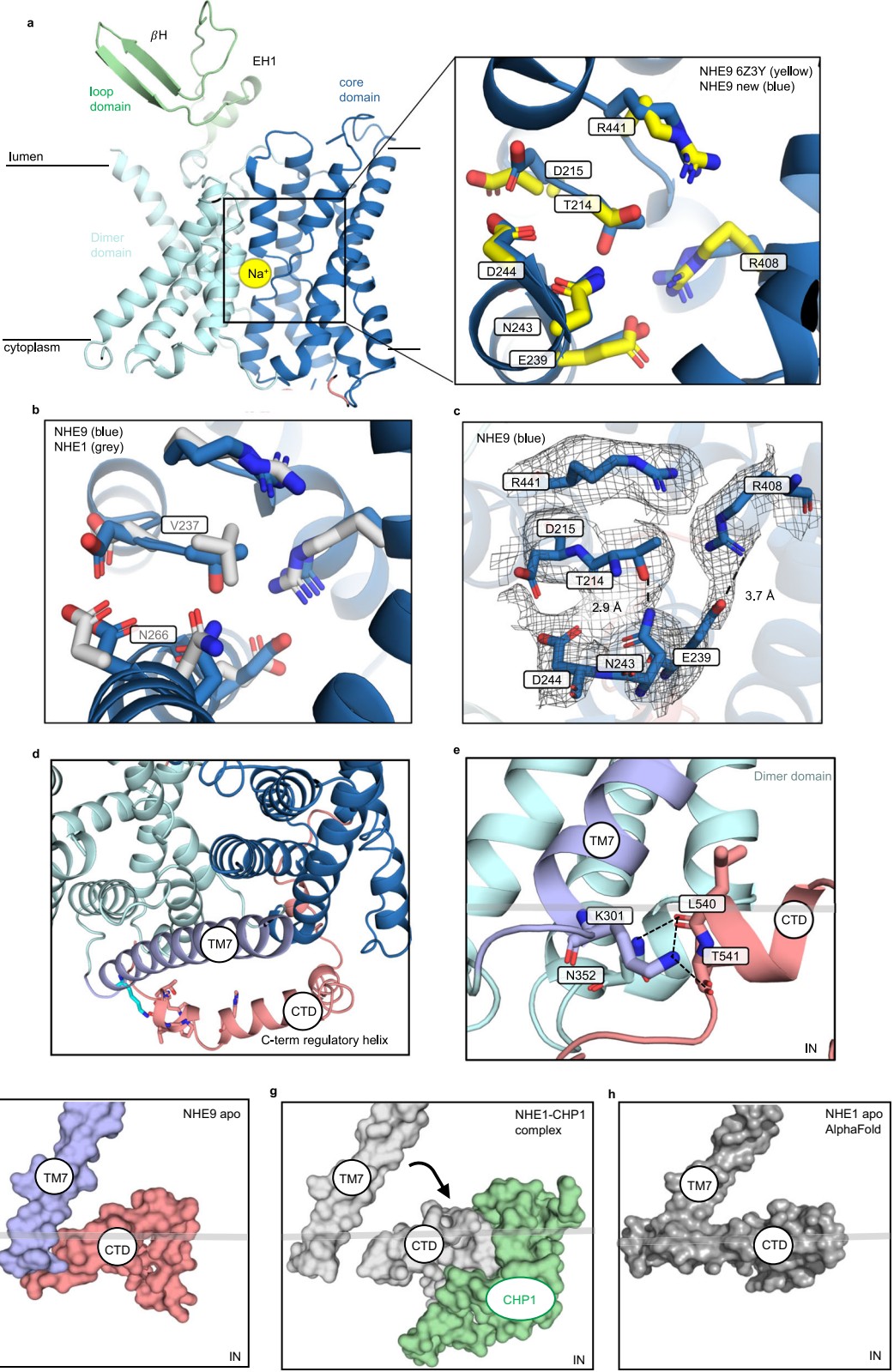

and set up 6 variable components with the filter resolution 4 Å and high-pass filter 20 Å, respectively. After cluster analysis, the remaining 103,815 particles in 1 of 3 cluster was selected and subsequently, homogeneous refinement in CryoSPARC was performed after applying C2 symmetry and the reconstructed map reached to 3.1 Å resolution at the gold standard FSC (0.143).

## Image processing NHE9*CC

The dataset was processed using CryoSPARC[34]. Dose fractionated movie frames were aligned using "patch motion correction," and contrast transfer function (CTF) were estimated using "Patch CTF estimation". The particles were picked using automated blob picker. Particles with good 2D classes were used for template based particle

**Fig. 6 | Ion-binding site and C-terminal domain of the NHE9*CC. a** *left:* cartoon representation of the NHE9 ion-binding site, located in the 6-TM core transporter domain, which is made up of two broken helices. The sodium ion (yellow sphere) is located and coordinated at the ion-binding site. *right:* ion-binding residues of the NHE9*CC structure are shown as blue sticks and labelled, with the identical residues of NHE9* shown as yellow sticks (PDB id: 6Z3Y). **b** Ion-binding residues of NHE9*CC are shown as blue sticks and NHE1 residues shown as grey sticks with Val237 and Asn266 labelled. **c** Hydrogen-bonding between T214-N243 and salt-bridge interaction between E239-R408 in the NHE9*CC structure are illustrated by dashed lines

and the cryo-EM maps shown as grey mesh. **d** Top view showing how the CTD helix is positioned on the outside of the mobile TM7 linker domain. **e** Numerous contacts are formed between the CTD and the linker helix TM7, in particular via K301 (blue-sticks) in TM7 to residues in the CTD helix (salmon sticks and labelled). **f** In the NHE9* CC structure the CTD stays connected to the linker helix TM7 (protein shown as surface). **g** CHP1 (green, surface) binding to the CTD of NHE1 (grey, surface) moves it away from the linker helix TM7. **h** Alfafold2 model of NHE1 and CTD interaction to TM7 predicted as was observed in the NHE9* CC cryo-EM structure.

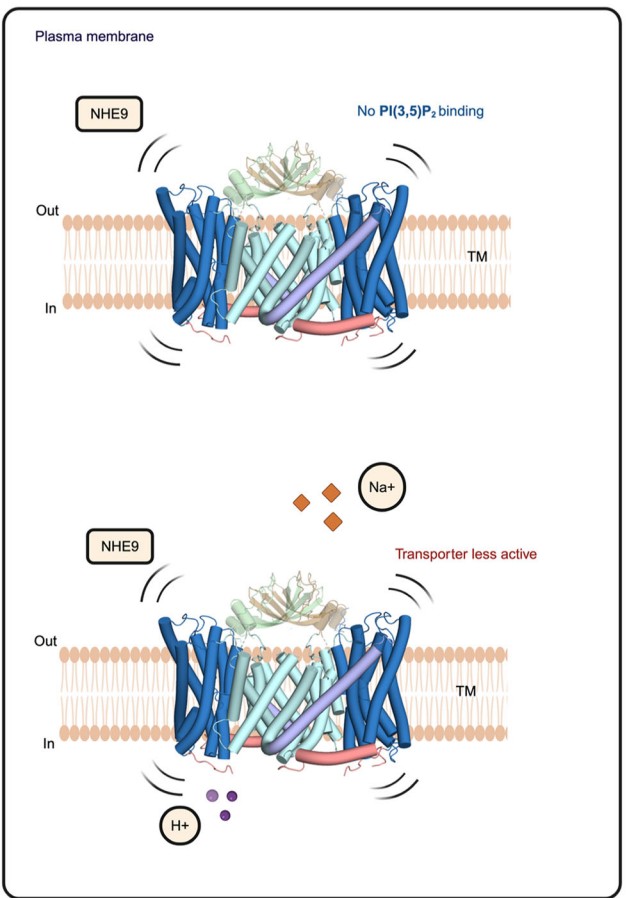
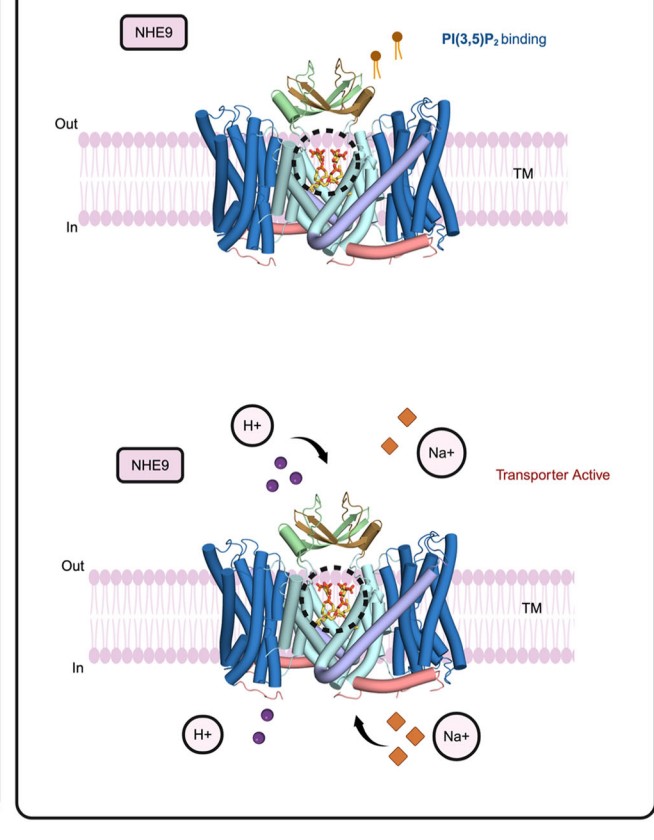

**Fig. 7 | The PI(3,5)P$_2$-dependent activation of NHE9 in endosomes.** Schematic representation of NHE9 ion translocation, dimer stabilisation and transport activation in the endosomes vs. the cell membrane. NHE9 is inactive in the plasma membrane. Upon relocation into endosomes, a concomitant binding of PI(3,5)P$_2$ at

the dimerization interface improves the stability of homodimer and activates the transporter. Created in BioRender. Kokane, S. (2025) https://BioRender.com/o88x200.

picking and extracted using a box size of 300 pixels. 268,054 particles were further used for ab initio model building and hetero refinements. Final round of non-uniform refinement with C2 symmetry and masked local refinement did result in 3D reconstruction with a gold standard FSC resolution estimation of 3.15 Å.

### Image processing EcNhaA-mut2
14,329 dose fractionated movie frames were aligned using "patch motion correction," and contrast transfer function (CTF) were estimated using "Patch CTF estimation" in CryoSPARC[34]. 4,401,429 particles were extracted and cleaned up using multiple rounds of 2D classification. 542,404 particles were used for ab initio model building and cleaned using multiple rounds of hetero refinement. 78,917 particles were further selected for a final round of non-uniform refinement and masked local refinement which resulted in 3D reconstruction with a gold standard FSC resolution estimation of 3.37 Å.

### Cryo-EM model building and refinement
Previously determined NHE9 structure (PDB id: 6Z3Z) was fitted into the cryoEM density map of NHE9* CC mutant. Iterative model building and real space refinement was performed using COOT[65] and PHENIX.refine[66]. Prior to model building of the extracellular loop domain, AlphaFold2 (AF2) with poly-glycine linker was performed to predict the NHE9 dimer structure assembly. After automatically fitting the initial model into the cryo-EM map, iterative model building and real space refinement was performed using COOT[65] and PHENIX.refine[66]. The refinement statistics are summarized in Supplementary Table 1. For generating figures illustrating structural features PyMOL[67] was used, and for figures including cryo-EM maps either Chimera[68] or ChimeraX[69] was used.

### All-atom molecular dynamics simulations
The NHE9* CC structure and NHE9* TM2-TM3 loop variant (K58Q, K105Q, K107Q) were embedded into the POPC bilayer and solvated in

0.15 M NaCl using CHARMM-GUI[70]. Either PI(3,5)P$_2$ and PI(4,5)P$_2$ were placed in the PIP$_2$ binding site identified from the cryo-EM structure. All simulations were simulated with a 2 fs timestep using the CHARMM36m forcefield in GROMACS 2022.1[71]. The system was then energy minimised and equilibrated using the standard CHARMM-GUI protocol[71], where the last step of the protocol was extended to 5 ns. The production runs were conducted for 250 ns under 303.15 K using the v-rescale thermostat[72]. The pressure of all systems was maintained at 1 bar using a C-rescale barostat[73]. All simulations were carried out in triplicates, where simulation frames were saved every 0.1 ns.

## GFP-based Thermal Shift assay

To characterize lipid binding for NHE9* we used the GFP-Thermal Shift assay[74]. In brief, purified NHE9*-GFP fusions was diluted in a [24] buffer containing 20 mM Tris, 150 mM NaCl and 0.03% (w/v) DDM 0.006% (w/v) CHS to a final concentration of 0.05–0.075 mg/ml with DDM added to a final concentration of 1% (w/v), and incubated for 30 min at 4 °C. Stock solutions of the respective lipids, phosphatidylinositol-bis-4,5-phosphate (PI(4,5)P$_2$ dioctanoyl Echelon Biosciences cat no. P-4508), and phosphatidylinositol-bis-3,5-phosphate (PI(3,5)P$_2$ dipalmitoyl, Echelon Biosciences cat no. P-3516) were prepared with final concentration of 1 mg/ml in buffer mentioned above. Respective lipids were added to purified NHE9* sample to a final concentration of 0.1 mg/ml and incubated for 10 mins on ice. Subsequently, β-D-Octyl glucoside (Anatrace) was added to a final concentration of 1% (w/v) in the sample aliquots of 100 µl were heated at individual temperatures ranging from 20–80 °C for 10 min using a PCR thermocycler (Veriti, Applied Biosystems). Heat-denatured material was pelleted at 5000 × g for 30 min at 4 °C. The resulting supernatants were collected and fluorescence values recorded (Excitation: 488, Emission 512 nm) using 96-well plate (Thermo Fisher Scientific) measured with Fluoroskan microplate fluorometer (Thermo Scientific) reader. The apparent $T$m was calculated by plotting the average GFP fluorescence intensity from three technical repeats per temperature and fitting a resulting curve using a sigmoidal 4-parameter logistic regression using GraphPad Prism software (GraphPad Software Inc.). The $\Delta T_M$ was calculated by subtracting the apparent $T_M$ of the lipid free sample, from the apparent $T_M$ of the sample with the respective lipid added.

For measuring lipid dependent dimer retention, the FSEC-TS protocol{Hattori, 2012 #274} was applied where the samples were heated at 50 °C for 10 min and pelleted as mentioned above. 80 µl of supernatant from each sample was injected onto an Enrich SEC 650 10 x 300 Column (BioRad) preequilibrated in buffer containing 20 mM Tris pH 7.5, 150 mM NaCl, and 0.03% (w/v) DDM for fluorescence detection size exclusion chromatography using a Shimadzu HPLC LC-20AD/RF-20A system. The dimer peak heights were normalized to NHE9* with PI(3,5)P$_2$ and the quantification determined from three independent FSEC experiments and plotted using Graphpad Prism (9.5) software

## Cell-based human NHE1 and NHE9 activity

Cells used for experiments were tested mycoplasma free and grown in a humidified 95/5% air/CO2-atmosphere incubator at 37 °C. PS120 fibroblasts, NHE1-null cells derived from Chinese hamster lung fibroblasts CCL39 cells[75], were cultured in high glucose DMEM supplemented with 10% fetal bovine serum (FBS), 100 units/mL penicillin and 100 µg/mL streptomycin (reagents from Sigma-Aldrich). Human full length NHE9 was cloned into the pMH vector (Roche) containing a C-terminal HA tag, human full length NHE1 was cloned into p3xFLAG-CMV-14 (Sigma-Aldrich) containing a C-terminal 3xFLAG tag. All constructs were verified by sequencing. Transient transfection was performed with Lipofectamine 2000 (Invitrogen) according to manufacturer's instructions, and experiments (cell surface expression and transport function studies) were performed 48 hrs after transfection.

## Cell surface biotinylation

Cell surface biotinylation was performed as described previously[76]. Cells were rinsed with 1x PBS and surface proteins were biotinylated by incubating cells with 1.5 mg/ml sulfo-NHS-LC-biotin in 10 mm triethanolamine (pH 7.4), 1 mm MgCl$_2$, 2 mm CaCl$_2$, and 150 mm NaCl for 90 min with horizontal motion at 4 °C. After labeling, plates were washed with quenching buffer (1xPBS containing 1 mm MgCl$_2$, 0.1 mm CaCl$_2$, and 100 mm glycine) for 20 min at 4 °C, then rinsed once with 1xPBS. Cells were then lysed in RIPA buffer (150 mm NaCl, 50 mm Tris·HCl (pH 7.4), 5 mm EDTA, 1% Triton X-100, 0.5% deoxycholate, and 0.1% SDS), and lysates were cleared by centrifugation. Cell lysates of equivalent amounts of protein were equilibrated overnight with streptavidin-agarose beads at 4 °C. Beads were washed sequentially with solutions A (50 mm Tris·HCl (pH 7.4), 100 mm NaCl, and 5 mm EDTA) three times, B (50 mm Tris·HCl (pH 7.4) and 500 mm NaCl) two times, and C (50 mm Tris·HCl, pH 7.4) once. Biotinylated proteins were then released by heating to 95 °C with 2.5× Lämmli buffer.

## SDS-PAGE, immunoblotting and antibodies

SDS-PAGE and immunoblotting were performed as described in ref. 77. Cell lysates were separated by SDS-PAGE and transferred to PVDF membranes. The membranes were incubated for 60 minutes with PBS-T20 containing 5% (w/v) skimmed milk. The membranes were then immunoblotted in 5% (w/v) skimmed milk in PBS-T20 with the indicated primary antibodies overnight at 4 °C. The blots were then washed three to six times with PBS-T20 and incubated for 1 h at room temperature with the appropriate secondary HRP-conjugated antibodies in 5% (w/v) skimmed milk in PBS-T20. After repeating the washing steps, the signal was detected with the enhanced chemiluminescence reagent. Immunoblots were developed using a film automatic processor (Fujifilm) and films were scanned with a 600 dpi resolution on a scanner and quantified with ImageJ software. Mouse monoclonal anti-FLAG M2 (Sigma Aldrich F3165), anti-HA (Sigma Aldrich H9658) and anti-Tubulin (Sigma Aldrich T9028) antibodies and were obtained and used at 1:1000 dilution. HRP-coupled goat anti-mouse IgG antibody (Invitrogen 31430) was used at 1:2000 dilution, avidin-HRP (Bio-Rad 1706528) was used at 1:1000 dilution.

## NHE1 and NHE9 transport activity

NHE activity was measured fluorometrically using the intracellularly trapped pH-sensitive dye BCECF (2′,7′-bis-(Carboxyethyl)-5(6′)-carboxyfluorescein Acetoxymethyl Ester) (Invitrogen B1170) with the NH$_4$Cl prepulse technique as described previously[76]. Cells grown on glass coverslips were loaded with 1 µM BCECF-AM and exposed to 25 mM NH$_4$Cl for 15 min in a buffer containing in mM: 120 NaCl, 5 KCl, 2 CaCl$_2$, 1.5 MgCl$_2$, 25 NH$_4$Cl, 30 Hepes titrated to pH 7.4 with N-methyl-D-glucamine (NMDG). Then, cells were washed 3x with and incubated in a buffer containing in mM: 120 Tetramethylammonium-Cl (TMACl), 5 KCl, 2 CaCl$_2$, 1.5 MgCl$_2$, 30 Hepes titrated to pH 7.4 with NMDG. Recording was started and after 60 s cells were rapidly exposed to a buffer containing in mM: 120 NaCl, 5 KCl, 2 CaCl$_2$, 1.5 MgCl$_2$, 30 Hepes titrated to pH 7.4 with NMDG. BCECF fluorescence signals (λ excitation: 490 and 440 nm, λ emission: 535 nm) were recorded in a computer-controlled spectrofluorometer (Fluoromax-2, Photon Technology International). The 490/440 nm fluorescence ratio was calibrated to pHi using the K$^+$/nigericin method, and initial rate (Vmax; ΔpH/Δtime) of sodium-dependent intracellular pH recovery calculated[76]. All steps of incubation, recording and calibration were performed at 37 °C.

## Measurement of endosomal pH

Fluorescence ratio imaging with pH-insensitive Alexa Fluor 633–transferrin (Invitrogen) and pH-sensitive pHrodo Red Transferrin Conjugates (Invitrogen) was performed to measure endosomal pH as described[78,79]. Cells were starved for 1 h at 37 °C in serum-free DMEM

and then incubated for 1 h at 37 °C in HBSS (in mM: 137 NaCl, 5.3 KCl, 1.3 Ca$_2$Cl, 0.82 MgSO$_4$, 0.34 Na$_2$HPO$_4$, 0.44 KH$_2$PO$_4$, 4.2 NaHCO$_3$, 5.6 glucose, and 15 HEPES–NaOH, pH 7.4) containing with 25 µg/mL pHrodo Red Transferrin Conjugates (Invitrogen) and 25 µg/mL Alexa Fluor 633–human transferrin (Invitrogen). After extensive washing of the cells, pHrodo™ Red and Alexa Fluor 633 fluorescence were acquired at 37 °C with a Leica SP8 confocal microscope equipped with a 63× oil immersion objective. A pH calibration curve was constructed by using a calibration solution containing 125 mM KCl, 25 mM NaCl, 10 µM nigericin, and one of the following buffers at a concentration of 25 mM: HEPES (pH 7.0) or MES (pH 6.5, 6.0 or 5.5). Fluorescence intensities of the perinuclear recycling endosomes were quantified with ImageJ software. In a single experiment, endosomal pH values of 30 randomly selected living cells were measured and averaged.

### Solid Supported Membrane-based electrophysiology

For SSM-based electrophysiology measurements, protein was reconstituted in yeast polar lipids (190001C-100 mg Avanti). The lipids were prepared by solubilization in chloroform and dried using a rotary evaporator (Hei-Vap Core, Heidolph Instruments). Dry yeast polar lipids were thoroughly resuspended in 10 mM MES-Tris pH 8.5, 10 mM MgCl$_2$ buffer at a final concentration of 10 mg ml$^{-1}$. Unilamellar vesicles were prepared by extruding the resuspended lipids using an extruder (Avestin) with 200-nm polycarbonate filters (#10417004 Nuclepore Track-Etch membrane). The vesicles were destabilized by the addition of Na-cholate (0.65% w/v final concentration). SEC-purified protein was added to the destabilized liposomes at a lipid-to-protein ratio (LPR) of 5:1 and incubated for 5 min at room temperature. The sample was added to a PD SpinTrap G-25 desalting column (Cytiva) for removing detergent and the reconstituted proteoliposomes were collected in a final volume of 100 µl. The sample was diluted to final lipid concentration of 5 mg ml$^{-1}$ in 10 mM MES-Tris pH 7.5, 10 mM MgCl$_2$ buffer, flash frozen in liquid nitrogen and stored at −80 °C until use. Proteoliposomes were diluted 1:1 (vol/vol) with non-activating buffer (10 mM MES-Tris pH 7.5, 300 mM Choline chloride, 10 mM MgCl$_2$) and sonicated using a bath sonicator. 10 µl of sample was loaded on 1 mm sensor (#161002 Nanion Technologies). For sample measured with PI(3,5)P$_2$, 20 µM of the lipid final (v/v) was added to the protein and incubated for 15 min. The sample was subsequently reconstituted in yeast polar lipids as described above.

Sensor preparation for SSM-based electrophysiology using the SURFE$^2$R N1(Nanion Technologies) system was performed as described previously[39]. During the experiments, NHE9* and NHE9* mutants were activated by solution exchange from non-activating buffer to an activating buffer containing the substrate i.e NaCl. For binding kinetics, $x$ mM Choline chloride was replaced by (300 - $x$) mM NaCl in the activating buffer at increasing concentrations. The Na$^+$-induced peak currents were fitted from triplicate measurements using nonlinear regression curve-fit analysis to a one site specific binding model using GraphPad Prism software. The peak current values were normalized with respect to the average of the maximum value obtained across all measurements. The final $K_d$ values reported are the mean±s.d. of $n$ = 3 independent sensors.

### Reporting summary

Further information on research design is available in the Nature Portfolio Reporting Summary linked to this article.

## Data availability

Data supporting the findings of this paper are available from the corresponding author upon request. The coordinates and the maps for *Ec*NhaA, NHE9* with loop, and NHE9*CC with PI(3,5)P$_2$ have been deposited in the Protein Data Bank (PDB) and Electron Microscopy Data Bank (EMDB) with entries PDB: 8PS0, EMD-17841; PDB: 8PXB, EMD-18002; PDB:8PVR, EMD-17971, respectively. Previously published

data for crystal structure of NhaA at pH 6.5, at pH 4, NHE9* and NHE1 are available with PDB codes PDB:7S24, PDB: 4AU5, PDB: 6Z3Z, PDB: 7DSW. The data for molecular dynamic simulation is available on Zenodo [https://zenodo.org/records/14679952]. Source data for Figs. 3a–e, 4c, e and f are provided with this paper in the Source data file. Source data are provided with this paper.

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

## Acknowledgements
We are grateful to Marta Carroni at the Cryo-EM Swedish National Facility at SciLifeLab Stockholm for cryo-EM data collection as well as Michael Hall at the Umeå Core Facility for Electron Microscopy, UCEM and the European Synchrotron Radiation Facility (ESRF). This work was principally funded by a European Research Council (ERC) Consolidator Grant EXCHANGE (grant #. ERC-CoG-820187) to D.D. It was further supported by the Swiss National Science Foundation (grant # 320030-231405) to D.G.F., The Knut and Alice Wallenberg Foundation to L.D.; The Science for Life Laboratory to L.D.; The Göran Gustafsson Foundation to L.D.; The Swedish eScience Research Center to L.D.; and The Swedish Research Council (grant #. VR 2019-02433 and 2022-04305) to L.D. The National Academic Infrastructure for Supercomputing in Sweden (NAISS) and the Swedish Research Council (grant #. 2022-06725) to L.D. funded MD simulations.

## Author contributions
D.D. designed the project. Cloning, expression screening and sample preparation for cryo-EM was carried out by P.F.M. and S.K. Cryo-EM data collection and map reconstruction was carried out by P.F.M, S.K, R.M, A.G and D.D. Model building was carried out by R.M, A.G and D.D. MD simulations were carried out by T.P and L.D. Cell-based pH measurements were carried out by G.A, and T. H and supervised by D.F. All authors discussed the results and commented on the manuscript.

## Funding

## Competing interests
The authors declare no competing interests.
