## [Transparent Peer Review file · Nature Communications]

PIP2-mediated oligomerization of the endosomal sodium/proton exchanger NHE9

Corresponding Author: Professor David Drew

Version 0:

Reviewer comments:

Reviewer #1

(Remarks to the Author)

This is a potentially impactful study which could broaden our understanding of the structure and mechanism of organellar Na⁺/H⁺ exchangers, which are generally less well studied compared to their plasma membrane counterparts. While the new structural information will move the field forward, the accompanying functional interpretation is less well justified. Major points are listed below.

1. Important new structural resolution reveals binding of lipid at the dimer interface: PI(3,5)P2 in NHE9 and cardiolipin in E. coli NHA2. The authors hypothesize that lipid binding confers scaffolding stability to the dimer and they provide in vitro data showing greater thermostabilization of a truncated form of horse NHE9 (NHE9*) in the presence of PI(3,5)P2. This study also focuses on the TM2-TM3 beta strand domain ("crown") which was not clearly resolved in an earlier study. Only organellar members of the eukaryotic NHE family appear to have this luminal domain.

(i) The introduction makes the argument that lipid binding is a feature of NHE members with weak dimer interface. It would be helpful for a more detailed Discussion justifying this potentially useful correlation. Perhaps summarize findings from available NHE structures relating to this point?

(ii) The functional connection between the crown domain and lipid binding is unclear. A triple Lys mutant in the crown abolishes dimer formation. It follows that PI lipids no longer have a stabilizing influence on the structure, since it is a monomer and the dimer interface is lost. Thus, these two functions may be only indirectly related? The model in Fig 6 shows the PI lipid inserting via the crown domain but I am not seeing any evidence to support this.

2. Why is K⁺ as a bound and transported ion never monitored in this study (Fig 3 and extended data Fig 8)? K⁺ transport is uniquely a feature of organellar NHE like NHE9 and likely to be the physiologically relevant ion within the cell.

3. "Consistent with a regulatory switch, PI(3,5)P2 is a minor lipid that is not found in the plasma membrane, but only found in late endosomes and lysosomes (34,35), which coincides with NHE9 localization."

(i) What is the evidence (reference) that NHE9 has been localized to late endosomes and lysosomes? Contrary to this statement, localization experiments actually suggest that NHE9 is excluded from late endosomes and lysosomes.

(ii) Have the authors considered the hypothesis that PI lipids downregulate transport activity? After all, this study does not measure pH/ion changes in endosomes, and only measures Na⁺ effects on structural stabilization which may not be linked to increased transport.

4. Fig 3c-d is problematic and lacking in controls.

(i) Why was the NHE9 construct switched from equine NHE9* used for all other experiments, to human NHE9? There are no data provided that this construct is functional. One obvious control would be to show that this transporter construct is functional in endosomes by measuring endosomal pH changes as reported by many groups in the field.

(ii) Both full length and the truncated construct of horse NHE9* should be evaluated in mammalian cells using pH based assays for physiological context with other experiments that use yeast-purified proteins.

(iii) No loading control in blot of 3c. How much of NHE9 makes it to the surface compared to NHE1? If only a small fraction, that could explain failure to see transport activity at the plasma membrane. Immunofluorescence staining of the cells should be shown for both transfectants.

(iv) Transport activity of organellar isoforms NHE6-9 at the plasma membrane has been reported by other groups (e.g., Milosavljevic et al., 2014) and is commonly used as a strategy to study these transporters. In light of this, the claim that plasma membrane localized NHE9 is inactive is intriguing (and potentially very important!) but deserves stronger justification. Do the authors believe the lack of activity observed at the plasma membrane in this study is unique to NHE9? If so, this would be important to demonstrate by expressing NHE6 and NHE7 in PS120 cells.

Reviewer #2

(Remarks to the Author)

The manuscript by Kokane et al. shows extensive structural, functional, and in-silico characterisations of the Na⁺/H⁺ exchangers' regulation by lipids, and is part of the long-term efforts in the Drew laboratory to understand the molecular mechanisms of these important transporters. The authors propose a novel and exciting mechanism for the regulation of the mammalian organellar NHE9 by PIP₂ lipids, whereby transport is selectively enhanced at the endosomal membranes through PI(3,5)P₂-mediated dimerization.

The manuscript is concise and clearly written. The data largely support the mechanistic claims, and the graphical part has been carefully made to represent the data. I am convinced that this work will be of great interest to organelle physiologists, and to the membrane transport field.

I have several comments that the authors might like to consider for the final version of the manuscript.

-Modelling of lipid molecules:

Without access to the EM maps, it is difficult to judge the level of certainty to assign specific lipid molecules to the EM maps at overall resolutions ~3Å. Indeed, it all boils down to the local quality of the map at the lipid binding sites, and the authors have been cautious with their statements about this in the text. However, from Fig 1b is difficult to see the density around the CDL molecules referred to in line 128 "...clear map density supports the modelling of cardiolipin..." (see typos below). To remediate this, Ext Dat Fig 2c could be improved (the EM density it is not smoothly contoured around the lipid molecules) and moved into Figure 1, so to show the EM density around the CDL molecules in the main part of the manuscript. Regarding the two bound PI(3,5)P₂ molecules modeled on the NH₉⁺-CC structure. It seems clear that the PI(3,5)P₂ headgroup fits better into the triangular density than that of the PI(4,5)P (Fig. 2b). However, the phosphate group at position 3 of one of the PI(3,5)P₂ bound molecules seems to be in very close proximity (within van der Waals maybe, Fig. 2c) to the phosphate group at position 5 of the second PI(3,5)P₂ molecule. How do the authors think that the negative charges of the opposing phosphate groups are shielded from one another to maintain their proximity? Do the basic residues (K105, K107) enable that? Water penetration? Alternatively, is it possible that only one PIP₂ molecule binds at a given time? Maybe, the MD simulation can shed light on this. Would it be possible to compute the distances between the opposing phosphate groups (3' in one PIP₂ molecule and 5' in the other one) during the simulations, and correlate that with the dynamics of the side chains of the lysine residues?

-Structure determination:

As the authors clearly stated, the dynamics of the dimer interface precluded high-resolution maps around the beta-hairpin and the bound lipid molecules. CryoSPARC was used with some success to disentangle that conformational heterogeneity. RELION can also be a powerful tool to do this, and I wonder if the authors attempted to 3D classify the particles using local masks encompassing the scaffold domains, and/or the lipid binding sites.

The authors speculate that the better cryoEM map quality of NHE9⁺CC over NH₉⁺ could be due to the mutations and/or the addition of lipids. However, the cys residues don't appear to be crosslinked (maybe the acidic pH contributed to that). One possible explanation for the map improvement -not discussed in the manuscript- is that the NHE9⁺CC dataset was collected in a different Krios setup with more a modern direct electron detector, and an energy filter.

-Functional assignment PIP₂ binding:

In Fig. 3a, the fit of the PI(4,5)P₂ data does not start from 100%, and I think it should. That would yield a slightly higher T_m. More importantly, both the thermal-shift and FSEC analyses show that PI(4,5)P₂ binds NH₉⁺ (Fig.3a,b), yet that lipid, as opposed to PI(3,5)P₂ doesn't have an effect on the apparent Na⁺ K_d measured by SSM. This is an interesting result, and it is not discussed in the manuscript. Moreover, the SSM experiments with PI(4,5)P₂ are important and should be included in Fig. 3, rather than in the Ext Dat Fig. 8. Please, avoid normalization of the Na⁺ titration curves in Ext Dat Fig. 8e, so that the effect of PI(4,5)P₂ on the amplitude of the capacitive transients, if any, can be appreciated. Displaying representative capacitive transients would also help in this regard. The error bars in some of the Na⁺ titrations plots are missing. Fig.3 also shows that NHE9, when expressed at the plasma membrane, does not give a transport signal in the fluorescence-based cell assay. Are the transport turnover rates of NHE9 and NHE1 known, and are they comparable? Could it be the turnover rate of NH₉ is much slower than that of NHE1 and that the fluorescence-based assay is not sensitive enough to measure NHE9 activity?

Finally, please consider using Na⁺ apparent binding affinity, instead of just affinity. Also, apparent melting temperature.

Further mechanistic insights:

Please, consider moving this part and the associated figure to a supplementary discussion, or alternatively attempt to connect it better with the effect of the PIP2 lipids on NHE function.

Typos:

-Line 128 "...clear map density to supports the modelling of cardiolipin..."

-In the methods, it is stated that the SEC buffer for the final purification step of NHE9*CC is at pH 7.5. Is this a typo?

-Line 251: "Without lipid addition, NHE9* unfolds around 30C with a shallow slope..." Figure 3a shows an apparent Tm of ~50C, as stated in the figure legend.

Reviewer #3

(Remarks to the Author)

Kokane et al. presents a interesting study outlining the lipid-dependent dimerization of the sodium-proton exchangers (NHE). They present three novel resolved structures of Na⁺/H⁺ exchangers (NHE) and outline the specific lipids that play a role in the dimerization interface of the proteins. Additionally, they outline the role for a β -hairpin domain that was previously unresolved in other structures. They also provide evidence for endosomal localization of NHE9. The overall manuscript is well written, but I would recommend publication of the manuscript given the following comments have been addressed:

Major comments:

1. The authors have resolved the structure of the WT-activity like mutant A109T, Q277G and L296M of NhaA. However, I don't see any activity data characterization for the triple mutant, with comparison to the Wild-Type activity. If there is a previous study highlighting the activity of this mutant, please cite. Otherwise, please present supporting evidence that the activity of the triple mutant is unaffected.

2. The authors mention that the specific lipid environment plays a critical role in regulating NHE9*, and that the endosomal lipidic environment is critical to NHE9* activity. However, the MD simulations performed in the manuscript don't use a complex membrane, but rather use a simplified membrane (POPC with the PIP2 lipid embedded) for their simulations. This feels counter-intuitive, and I suppose the results would be better explained by simulations performed in a realistic membrane environment.

3. Additionally, the simulations seem too short for the overall system (250 ns), since it has been shown that membrane proteins take a much longer time to equilibrate (~0.5 - 1 microseconds)

4. There are many interesting experimental observations reported in this study which could have benefited by the physical insights obtained from the MD simulations. However, authors have chosen a narrow question related to mutations in charged residues to obtain an expected results. MD simulations section of the manuscript could have been improved with better design of questions for the computational investigation.

Minor comments:

The title could mention NhaA, since the manuscript spends a significant time discussing the NhaA structure.

Abstract:

Line 25: 'all' cells? Could the authors be more specific?

Line 31: dependent on of the lipid cardiolipin. The authors use "horse" throughout the paper. Could they replace it with the scientific name (*E. caballus*)?

Lines 38-41: Break the long sentence into multiple sentences to improve readability.

Introduction:

Lines 75-76: Citation missing

Line 90, 93: Citation style changes mid paragraph.

Results:

Lines 107-114: The authors don't mention the length of the overall resolved structure (which residues were successfully modelled, etc.)

Line 110: Mutation style is inconsistent: (A109T on line 11), while the rest of the manuscript uses three-letter mutations. I would personally recommend to change all three-letter names to single letter, to improve readability.

Line 161: Citation missing

Line 165: Extended Data Fig. 4a instead of 3a?

Line 183: Could the authors present the 60° angle in Fig. 2?

Line 184: confidently?

Line 186: ECH1 is not labeled in Fig.2

Line 196: Is NHE9*CC resolved at the same pH as NHE9*? Please clarify.

Lines 328-329: The RMSD < 3Å does not necessarily imply that the lipid stayed < 3Å from its initial position. Please correct. What software were used to compute the RMSD, and contact probabilities? Citations are missing. Could the authors use p-values to compare the distributions in Fig. 4b-e?

Line 345: confidently?

Line 351: phylogenetic or phylogenetic?

Lines 361-362: Could the authors further elaborate on the conclusion? The explanation for the promiscuity of the endosomal transporters seems incomplete.

Line 424: There Their

Line 627: Wrong citation for CHARMM-GUI, it should be the all-atom citation instead of the coarse-grained

Version 1:

Reviewer comments:

Reviewer #2

(Remarks to the Author)

The authors have addressed all my concerns, and I congratulate them on their work.

Reviewer #3

(Remarks to the Author)

All comments raised in the first round of review have been addressed. I congratulate the authors on this excellent article.

We thank all referees for their considered evaluation. We have responded as appropriate to all queries below.

Reviewer #1 (Remarks to the Author):

This is a potentially impactful study which could broaden our understanding of the structure and mechanism of organellar Na^+/H^+ exchangers, which are generally less well studied compared to their plasma membrane counterparts. While the new structural information will move the field forward, the accompanying functional interpretation is less well justified. Major points are listed below.

1. Important new structural resolution reveals binding of lipid at the dimer interface: PI(3,5)P2 in NHE9 and cardiolipin in *E. coli* NhaA. The authors hypothesize that lipid binding confers scaffolding stability to the dimer and they provide in vitro data showing greater thermostabilization of a truncated form of horse NHE9 (NHE9*) in the presence of PI(3,5)P2. This study also focuses on the TM2-TM3 beta strand domain (“crown”) which was not clearly resolved in an earlier study. Only organellar members of the eukaryotic NHE family appear to have this luminal domain.

- (i) The introduction makes the argument that lipid binding is a feature of NHE members with weak dimer interface. It would be helpful for a more detailed Discussion justifying this potentially useful correlation. Perhaps summarize findings from available NHE structures relating to this point?

Thanks for raising this point, which could have been clearer. In the introduction we discuss the differences between the bacterial Na^+/H^+ exchangers NhaA and NapA. Although both essentially carry out the same reaction ($2\text{H}^+ : 1\text{Na}^+$) with similar kinetics, the activity of NhaA is dependent on cardiolipin, whereas the activity of NapA is not. The main structural difference is that NhaA has one less helix and forms a weaker oligomeric interface via β -hairpin domains now located in the scaffold domain. In contrast, NapA forms stronger protein-protein mediated oligomerization by its additional N-terminal helix. The differences between these two proteins matches a general observation that membrane proteins with weaker oligomerization interfaces tend to be more lipid-dependent¹. Indeed, NhaA is non-functional and forms a weaker dimer in an *E. coli* strain where the gene for cardiolipin synthesis has been deleted².

Extending this principle to other Na^+/H^+ exchangers we highlighted in the introduction that NHA2 has yet another N-terminal helix to NapA and this again alters the oligomeric interface. For NHA2 the oligomeric interface is also fairly weak and just two point mutations are enough to shift a large population of the protein to a monomer upon extraction in a mild detergent³. Indeed, NHA2 monomer mutants are no longer able to complement growth in a salt-sensitive yeast strain, although they remain well-folded³.

The NhaA (12 TMs) and NHA2 (14 TMs) are examples that show the most dramatic differences in the structures are located in the scaffold domain, as they have a different number of helices than the NHEs (13 TMs) and this makes their oligomerization both weaker and more lipid-dependent. With this context in place, the question is whether the NHE (13 TM) members follow the same trend or not?

Both yes and no. The size of the interface is 1/3 smaller for NHE9 than for NHE1 and NHE3 proteins and so this follows the same trend, but the interface for NHE9 is also larger than NapA. However, while the NapA interface is smaller, there is no gap between the protomers and so the dimer is more stable in NapA. In contrast, NHE9 has a larger gap between the protomers, which makes the interface more dynamic as lipids and detergents can bind here to influence stability. In addition, the β -hairpin domains in NHE6, NHE7 and NHE9 are located on the same helices that form dimerization contacts. Because the β -hairpins are so dynamic (they are difficult to build) we think their addition destabilizes the dimer interface. In other words, the presence of the β -hairpins makes oligomerization more susceptible to the requirement of lipid binding.

(ii) The functional connection between the crown domain and lipid binding is unclear. A triple Lys mutant in the crown abolishes dimer formation. It follows that PI lipids no longer have a stabilizing influence on the structure, since it is a monomer and the dimer interface is lost. Thus, these two functions may be only indirectly related? The model in Fig 6 shows the PI lipid inserting via the crown domain but I am not seeing any evidence to support this.

We were trying to make a simplified schematic, but in doing so we conveyed the incorrect message. We think the most likely scenario is that in the absence of PI(3,5)P₂ NHE9 has a weakened dimer interface, which would be either “less” active or “inactive”. As shown here, the PI(3,5)P₂ lipid is coordinated by the helices at the scaffold interface, which then recruits the β-hairpin domains and the lysine residues to further stabilize the bound lipid and the dimer. Indeed, the β-hairpin and the linker region are very flexible and could very well adopt a more open conformation in the absence of the lipid.

In support of this model, detergent extraction of the triple lysine mutant has both homodimer and monomer forms prior to purification. After purification in detergent all of the protein shifted to the monomeric form, even when purified in detergent with additional brain lipids added. Likewise, the NHE9* protein also breaks down into monomers “if” purified without brain lipids added in the detergent during wash steps. Thus, whilst why the PI(3,5)P₂ lipid helps to stabilize the homodimer in detergent it’s not strictly required for dimerization/folding per se. The same is seen for NhaA when expressed in E. coli absence of cardiolipin. Although its dimeric in membranes it quickly breaks apart into monomers upon detergent extraction.

We have now updated the schematic to better reflect our model for NHE9 activation.

2. Why is K^+ as a bound and transported ion never monitored in this study (Fig 3 and extended data Fig 8)? K^+ transport is uniquely a feature of organellar NHE like NHE9 and likely to be the physiologically relevant ion within the cell.

We tested ion specificity of NHE9, but here our focus was on its allosteric regulation. NHE9 also binds K^+ , yet with poorer affinity than Na^+ . Currently we are preparing a manuscript on the structure and function of human NHE6 where we have a larger focus on ion specificity. We would prefer to publish this data later with this study.

Fit of transient currents recorded for NHE9*(black) as a function of K^+ concentrations and the corresponding binding affinity across two different sensors measured thrice. The traces for empty liposomes(grey) were recorded as control.

3. “Consistent with a regulatory switch, PI(3,5)P2 is a minor lipid that is not found in the plasma membrane, but only found in late endosomes and lysosomes (34,35), which coincides with NHE9 localization.”

(i) What is the evidence (reference) that NHE9 has been localized to late endosomes and lysosomes? Contrary to this statement, localization experiments actually suggest that NHE9 is excluded from late endosomes and lysosomes.

We thank the reviewers for pointing this out. We meant that PI(3,5)P2 is present in late endosomes and lysosomes and the NHE9 localization overlaps with the localization to late endosomes, i.e., there is no evidence NHE9 is in lysosomes.

The paper we referred to has monitored multiple NHEs at the same time and found that NHE9 was located differently than NHE6 and NHE7, and was located in late endosomes⁴. The consensus location is NHE9 is at least in recycling endosomes⁵, which is in agreement with our own observations and clearly NHE9 will transit through early endosomes even if NHE6 is the main Na^+/H^+ exchanger in early endosomes.

Working with Michael Landreh / Carol Robinson (Oxford University) we have shown by native MS that NHE9* natively retains lipids matching the mass of PIP2 and from our new analysis here can conclude this lipid is PI(3,5)P2. Although we can't exclude other lipids binds to NHE9 at the dimer interface, clearly NHE9 prefers to bind a lipid that is present in much smaller quantities than most lipids.

The kinase for converting PI3P to PI(3,5)P2 is also in early endosomes and so PI(3,5)P2 is also present in early endosomes^{6,7}. Given the uncertainties in the exact NHE9 localization and the distribution of the PI(3,5)P2 lipids, we think it is most prudent to state that NHE9 is expressed in endosomes and that the PI(3,5)P2 is an endosomal lipid, rather than specifying the exact endosome locations.

ii) Have the authors considered the hypothesis that PI lipids downregulate transport activity? After all, this study does not measure pH/ion changes in endosomes, and only measures Na⁺ effects on structural stabilization which may not be linked to increased transport.

Yes, we have considered this, but all the evidence we have from a number of different Na⁺/H⁺ exchangers is that the dimer form is the active state of the protein. Since PI lipids stabilise the dimer it would make most sense that the lipid activates the protein. It's the same for CDL binding to NhaA, and for this protein we have clear in vivo evidence that CDL is required for NhaA activity²; this is one of the reasons why we wanted to combine NhaA and NHE9 structures in the same paper. In NHE9, we think the β -hairpin domain is not critical for coordinating the lipid per se, but it stabilises the lipid once it is bound. So why have the β -hairpin domain at all? Well the outside surface of the β -hairpin domain is negatively-charged and so think that by stabilizing this domain we create a larger attractive surface for cation binding.

4. Fig 3c-d is problematic and lacking in controls.

(i) Why was the NHE9 construct switched from equine NHE9* used for all other experiments, to human NHE9? There are no data provided that this construct is functional. One obvious control would be to show that this transporter construct is functional in endosomes by measuring endosomal pH changes as reported by many groups in the field.

There are only a few minor differences between horse and human NHE9 and the amino acid differences are in peripheral loop regions and none are close to the lipid-binding domain or the active site. Because all other studies have used human NHE9 and it has been (already) shown to be active in endosomes we thought we should also work with the human NHE9 protein for in vivo studies.

	1	10	20	30	40	50	60
humanNHE9	MERCS	VMSKEDYQ	QHQGAVELLV	FNFLIL	LTLT	IWLFKNHR	FRFLHETGGAMVYGL
horseNHE9	MERCR	VMSKEDYQ	QHQGAVELLV	FNFLIL	LTLT	IWLFKNHR	FRFLHETGGAMVYGL
	70	80	90	100	110	120	
humanNHE9	IMGLILRYATAP	TDIESGTVYDC	VKL	TFSPSTLL	VNI	TDQVVEYKYKREISQHNINP	HQCG
horseNHE9	IMGLILRYATAP	TDIESGTVYDC	GKLA	TFSPSTLL	VNI	TDQVVEYKYKREISQHNINP	HILG
	130	140	150	160	170	180	
humanNHE9	NAILEKMTFDPE	IFFNVLLPPI	IIFHAGYSLKRRHFF	QNLGSI	LTYAF	LGTAISCIVIGLI	
horseNHE9	NAILEKMTFDPE	IFFNVLLPPI	IIFHAGYSLKRRHFF	QNLGSI	LTYAF	LGTAISCIVIGLI	
	190	200	210	220	230	240	
humanNHE9	MYGFVKAM	YHAGQLKNGDF	HFTDCLFF	FGSLMSATDP	VTVLAIF	HELVDPDLYTL	LLFGES
horseNHE9	MYGFVKAM	YVAGQLKNGDF	HFTDCLFF	FGSLMSATDP	VTVLAIF	HELVDPDLYTL	LLFGES
	250	260	270	280	290	300	
humanNHE9	VLNDAVAIVL	YSISIYSPKENP	NAFDAAAF	QSVGNFL	CGIFAGSFAMGSAYA	IITALL	L
horseNHE9	VLNDAVAIVL	YSISIYSPKENP	NAFDAAAF	QSVGNFL	CGIFAGSFAMGSAYA	VVITALL	L
	310	320	330	340	350	360	
humanNHE9	RF	TKLCEFFMLE	TGLFFLLSWSA	FLSAEAAAGLT	GIVAVLFC	GVTAQAHYTYNNLS	SDSRIR
horseNHE9	RF	TKLCEFFMLE	TGLFFLLSWSA	FLSAEAAAGLT	GIVAVLFC	GVTAQAHYTYNNLS	LSDSRMR
	370	380	390	400	410	420	
humanNHE9	TKQL	FEFNF	LAENVIFCYMGLAL	FTFQNHIF	NALFILGAP	LAIFVARACNIYPLS	FLLN
horseNHE9	TKQL	FEFNF	LAENVIFCYMGLAL	FTFQNHIF	NALFILGAP	LAIFVARACNIYPLS	FLLN
	430	440	450	460	470	480	
humanNHE9	LGRK	QKIPWNFQ	HMMMFSG	LRGATAFALAI	RNTESQPKOMMF	ITLLLVFF	TVVVF
horseNHE9	LGRK	HKIPWNFQ	HMMMFSG	LRGATAFALAI	RNTESQPKOMMF	ITLLLVFF	TVVVF
	490	500	510	520	530	540	
humanNHE9	TPML	TWLQIRV	GDVLDENL	KEDEPSSQ	HOEANNL	DKNMTKA	ESARLFRMWYS
horseNHE9	TPML	TWLQIRV	GDVLDENL	KEDEPSSQ	HOEANNL	DKNMTKA	ESARLFRMWYGF
	550	560	570	580	590	600	
humanNHE9	L	THSGPPL	TTTTLPEW	CGPISRLL	TSPQAYGEQL	KEDDVE	CIVNQDELA
horseNHE9	L	THSGPPL	TTTTLPEW	CGPISRLL	TSPQAYGEQL	KEDDVE	CIVNQDELA
	610	620	630	640			
humanNHE9	P	PARLGLDQKA	S	PQTPGK	ENIYEGDLGLGGYELKLEQT	LCOSQLN	
horseNHE9	P	PARLGLDQKA	S	PQTPGK	ENIYEGDLGLGGYELKLEQT	LCOSQLN	

Sequence alignment of *hsNHE9* and *eqNhe9* having 95% sequence identity.

We have now shown that the *human* NHE9 construct we used is also active in endosomes

Recycling endosome pH measurement of PS120 cells transfected with empty vector (EV; negative control) or human NHE9. Data are shown as individual observations with mean \pm SD and represent three individual experiments combined. Each dot represents measurement of an individual cell. Asterisk denotes significance for the indicated comparison (two-tailed unpaired Student's t-test; **** $p < 0.0001$). pH calibration curve for endosomal pH measurements depicted in Fig. 5. For details see Methods section.

(ii) Both full length and the truncated construct of horse NHE9* should be evaluated in mammalian cells using pH based assays for physiological context with other experiments that use yeast-purified proteins.

We have carried out *in vitro* experiments for eqNHE9 as this protein was easier to produce in yeast to higher levels than *human* NHE9. Given all the residues for ion-binding and lipid coordination are the same between these two species, then we do not think its needed to repeat these experiments on the *horse* NHE9 isoform too. Indeed the AF2 prediction between eqNHE9 and human NHE9 is virtually the same.

(iii) No loading control in blot of 3c. How much of NHE9 makes it to the surface compared to NHE1? If only a small fraction, that could explain failure to see transport activity at the plasma membrane. Immunofluorescence staining of the cells should be shown for both transfectants.

All the controls have now been included

a. Plasma membrane expression of FLAG-tagged NHE1 and HA-tagged NHE9 in PS120 cells. PS120 were transfected with empty vector (EV; pMH), human NHE1 or human NHE9. Plasma membrane proteins were isolated using surface biotinylation. Avidin–HRP was used as loading control for surface proteins. The absence of a strong tubulin signal confirms the purity of the membrane fraction. Three independent biological replicates per condition. **b.** Immunoblots of PS120 cell lysates depicted in fig **a**. Three independent biological replicates per condition. **c.** Immunoblots of PS120 cell lysates and biotin fractions for FLAG-tagged NHE1 (left panel) and HA-tagged NHE9 (right panel). Indicated is the percentage of the input loaded on the SDS-page gel. Three independent biological replicates per condition. **d.** NHE transport activity in transfected PS120 cells. Measurement of plasma membrane NHE transport activity in empty vector (EV; negative control), human NHE1 (positive control) or human NHE9 transfected PS120 cells by quantification of the sodium-dependent intracellular pH recovery after acidification of the cytoplasm. Values are shown as means \pm SD.

(iv) Transport activity of organellar isoforms NHE6-9 at the plasma membrane has been reported by other groups (e.g., Milosavljevic et al., 2014) and is commonly used as a strategy to study these transporters. In light of this, the claim that plasma membrane localized NHE9 is inactive is intriguing (and potentially very important!) but deserves stronger justification. Do the authors believe the lack of activity observed at the plasma membrane in this study is unique to NHE9? If so, this would be important to demonstrate by expressing NHE6 and NHE7 in PS120 cells.

The only report we are aware of that shows transport activity of an organellar isoform at the plasma membrane is the study you have referred too is for NHE7⁸. Here, they have reported a very high K_M for Na^+ at 240 mM, which is in contrast to the 10-20 mM K_M reported for NHE1-NHE3 isoforms using either $^{22}\text{Na}^+$ uptake and fluorometry^{9,10}. Our K_D measurements for the NHE9 loop (monomeric) mutant with PI(3,5)P2 by SSM is 80 mM, but around 10 mM for the NHE9* protein with PI(3,5)P2 addition. We think although some activity for the organelle

NHE7-NHE9 isoforms may be measurable at the plasma membrane, this is very poor activity compared to the activity in endosomes.

Yes, in our hands *human* NHE6 is also inactive at the plasma membrane, but we are currently working on the manuscript of the cryo EM structure of NHE6 that shows that shows how lipids are also bound at the interface and that in the absence of these lipids that the homodimer is unstable. While it may seem that structural work is more routine with cryo EM, it still taken us a few years to obtain human NHE6 structures. Moreover, for NHE6 we would like to focus on other aspects, such as the mutations causing Christianson syndrome. Thus, while complementary, we think the current paper is already fairly extensive and I would rather include this data on a paper focused on NHE6.

Reviewer #2 (Remarks to the Author):

The manuscript by Kokane et al. shows extensive structural, functional, and in-silico characterisations of the Na⁺/H⁺ exchangers' regulation by lipids, and is part of the long-term efforts in the Drew laboratory to understand the molecular mechanisms of these important transporters. The authors propose a novel and exciting mechanism for the regulation of the mammalian organellar NHE9 by PIP2 lipids, whereby transport is selectively enhanced at the endosomal membranes through PI(3,5)P2-mediated dimerization. The manuscript is concise and clearly written. The data largely support the mechanistic claims, and the graphical part has been carefully made to represent the data. I am convinced that this work will be of great interest to organelle physiologists, and to the membrane transport field.

I have several comments that the authors might like to consider for the final version of the manuscript.

-Modelling of lipid molecules:

Without access to the EM maps, it is difficult to judge the level of certainty to assign specific lipid molecules to the EM maps at overall resolutions ~3Å. Indeed, it all boils down to the local quality of the map at the lipid binding sites, and the authors have been cautious with their statements about this in the text. However, from Fig 1b is difficult to see the density around the CDL molecules referred to in line 128 "...clear map density supports the modelling of cardiolipin..." (see typos below). To remediate this, Ext Dat Fig 2c could be improved (the EM density it is not smoothly contoured around the lipid molecules) and moved into Figure 1, so to show the EM density around the CDL molecules in the main part of the manuscript.

Thank you and apologies for not being clear here. We have rendered additional figures for the lipids. The local resolution estimates for CDL are similar to that of the overall protein estimates at 3.2 Å resolution. For bound lipids these are excellent cryo EM maps and I can't think of any other lipid fitting in well here, but still I think it's worth pointing out that we have modelled CDL based on all the other analysis we and others had done previously pointing to the role of CDL and dimerization.

Regarding the two bound PI(3,5)P2 molecules modeled on the NHE9*-CC structure. It seems clear that the PI(3,5)P2 headgroup fits better into the triangular density than that of the PI(4,5)P

(Fig. 2b). However, the phosphate group at position 3 of one of the PI(3,5)P₂ bound molecules seems to be in very close proximity (within van der Waals maybe, Fig. 2c) to the phosphate group at position 5 of the second PI(3,5)P₂ molecule. How do the authors think that the negative charges of the opposing phosphate groups are shielded from one another to maintain their proximity? Do the basic residues (K105, K107) enable that? Water penetration? Alternatively, is it possible that only one PIP₂ molecule binds at a given time? Maybe, the MD simulation can shed light on this. Would it be possible to compute the distances between the opposing phosphate groups (3' in one PIP₂ molecule and 5' in the other one) during the simulations, and correlate that with the dynamics of the side chains of the lysine residues?

These are excellent points. We have pasted the previous native MS analysis of NHE9*.

Although the native MS is an average of many different NHE9 proteins, the two largest peaks are of similar intensity and so its most likely that that NHE9 homodimer co-purifies with two PIP₂ lipids. We also know that the triple lysine mutant does not and so these should be the lipids at the homodimer interface. In the current maps the pocket is well solvated by water and so we don't think it's a problem for modelling this way and they were stable when refined in phenix and in MD simulations. However, we know that we need the lysine residues in the β -hairpin domain to stabilise these bound lipids. So we think that the flexible β -hairpin domains can adjust their position to help in the coordination.

To answer the reviewer question on how both phosphates are able to accommodate the same site without repelling each other, we computed the distances between each phosphate group and the lysine residues. Our simulation shows that the P3 phosphate of the first PI(35)P₂ is usually not coordinated with lysine. However, the P5 phosphate is usually either coordinated

with K105 on the first subunit (repeat 1,3), or the second subunit (repeat 2). On the second PI(3,5)P2 molecule, the P3 phosphate is coordinated with K107 on the second subunit of NHE9, whilst the P5 phosphate is coordinated with K105, which can be from either subunit. These co-ordination between multiple lysines help neutralize the negative charge within the phosphate headgroup and hence, allow two PI(35)P2 molecules to remain in close proximity to each other (Supplementary figure 9b).

-Structure determination:

As the authors clearly stated, the dynamics of the dimer interface precluded high-resolution maps around the beta-hairpin and the bound lipid molecules. CryoSPARC was used with some success to disentangle that conformational heterogeneity. RELION can also be a powerful tool to do this, and I wonder if the authors attempted to 3D classify the particles using local masks encompassing the scaffold domains, and/or the lipid binding sites.

We ran multiple rounds of 3D heterogeneous refinement and also 3D classifications without alignment using local masks encompassing the scaffold domains in cryosparc. Unfortunately, we were unable to improve the density of lipids and loop domain.

The authors speculate that the better cryoEM map quality of NHE9*CC over NHE9* could be due to the mutations and/or the addition of lipids. However, the cys residues don't appear to be crosslinked (maybe the acidic pH contributed to that). One possible explanation for the map improvement -not discussed in the manuscript- is that the NHE9*CC dataset was collected in a different Krios setup with more a modern direct electron detector, and an energy filter.

Yes, we agree that the Nhe9*CC dataset was collected on a slightly better detector and also a larger dataset could be collected because of the more modern setup, which could have partially resulted in the improved resolution. But we also think that the observed map improvement could be due to the addition of lipids since we could also observe density for the CTD helix, which directly interacts with the linker helix and scaffold domain and potentially affects the dynamicity of the transporter.

-Functional assignment PIP2 binding:

In Fig. 3a, the fit of the PI(4,5)P2 data does not start from 100%, and I think it should. That would yield a slightly higher Tm. More importantly, both the thermal-shift and FSEC analyses show that PI(4,5)P2 binds NHE9* (Fig.3a,b), yet that lipid, as opposed to PI(3,5)P2 doesn't have an effect on the apparent Na⁺ Kd measured by SSM. This is an interesting result, and it is not discussed in the manuscript. Moreover, the SSM experiments with PI(4,5)P2 are important and should be included in Fig. 3, rather than in the Ext Dat Fig. 8. Please, avoid normalization of the Na⁺ titration curves in Ext Dat Fig. 8e, so that the effect of PI(4,5)P2 on the amplitude of the capacitive transients, if any, can be appreciated. Displaying representative capacitive transients would also help in this regard. The error bars in some of the Na⁺ titrations plots are missing.

We thank the reviewers for pointing this out. We have the updated curves now which start from 100%. There are no differences in the T_M estimates. The plots are updated with error bars (we had included the data, but missed we had used the wrong function in prism and so they were not plotted). We have also shown the Na^+ -SSM traces prior to normalization.

Yes, we probably should have included a rationale for the stability also seen with the PI(4,5)P2 lipid addition. There are two explanations. The first one is that while we have measured the relative propensity for thermostabilization we haven't actually quantified lipid affinities. I don't know how would one do this for a lipid. In our assay, we saturate the system with a large excess of lipids as we measure a difference upon the resistance to heat-induced destabilization in detergent. So we can conclude that PI(3,5)P2 is relatively better at binding to NHE9 than PI(4,5)P2 under the same conditions. If we were to step-wise reduce the amount of lipid added, I presume we could find a concentration where only PI(3,5)P2 stabilizes the protein. In an in vivo situation it might be that only PI(3,5)P2 can bind effectively, and so in liposome assays where other lipids can start to interact with the binding site, we may only see an increase in activity for PI(3,5)P2 and not PI(4,5)P2 addition.

Another possible explanation is that upon heating we expose positive-charges that can non-specifically interact with the PIP2 lipids and so part of the thermostabilisation observed could be non-specific. What is important to note here, however, is the shape of the melting curve, which is different for PI(3,5)P2 than PI(4,5)P2. In particular, with PI(3,5)P2 we see a sharper transition, rather than a broader melting transition for PI(4,5)P2. This is a clear sign that PI(3,5)P2 lipid better stabilizes a single conformation of the protein, i.e., the homodimer. In protein unfolding studies the shape (slope) of the unfolding curve is just as important as the T_M estimate.

Fig.3 also shows that NHE9, when expressed at the plasma membrane, does not give a transport signal in the fluorescence-based cell assay. Are the transport turnover rates of NHE9 and NHE1 known, and are they comparable? Could it be the turnover rate of NHE9 is much slower than that of NHE1 and that the fluorescence-based assay is not sensitive enough to measure NHE9 activity?

While we can't exclude this possibility, our measured K_M for NHE9 in liposomes is similar to the K_M estimates for NHE1. We now show that we can detect NHE9 activity in endosomes, as should be expected and so we can confirm that there is nothing wrong with the setup per se.

Finally, please consider using Na⁺ apparent binding affinity, instead of just affinity. Also, apparent melting temperature.

Yes, good point. We see this as an apparent affinity and have updated the Ms accordingly.

Further mechanistic insights:

Please, consider moving this part and the associated figure to a supplementary discussion, or alternatively attempt to connect it better with the effect of the PIP2 lipids on NHE function.

Good point. We have now removed this section and incorporated the most important points into the discussion.

Typos:

-Line 128 "...clear map density to supports the modelling of cardiolipin..."

The statement has been changed

-In the methods, it is stated that the SEC buffer for the final purification step of NHE9*CC is at pH 7.5. Is this a typo?

Yes, the mentioned buffer conditions are changed. The purification for NHE9*CC was done at pH 8. We have changed the method section accordingly.

-Line 251: "Without lipid addition, NHE9* unfolds around 30C with a shallow slope..." Figure 3a shows an apparent T_m of ~50C, as stated in the figure legend.

30°C would be the temperature at which the NHE9*dimer starts unfolding. Apparent T_m of 50°C represents the temperature at which 50% population of NHE9*protein is unfolded.

Reviewer #3 (Remarks to the Author):

Kokane et al. presents a interesting study outlining the lipid-dependent dimerization of the sodium-proton exchangers (NHE). They present three novel resolved structures of Na⁺/H⁺ exchangers (NHE) and outline the specific lipids that play a role in the dimerization interface of the proteins. Additionally, they outline the role for a β-hairpin domain that was previously unresolved in other structures. They also provide evidence for endosomal localization of NHE9. The overall manuscript is well written, but I would recommend publication of the manuscript given the following comments have been addressed:

Major comments:

1. The authors have resolved the structure of the WT-activity like mutant A109T, Q277G and L296M of NhaA. However, I don't see any activity data characterization for the triple mutant, with comparison to the Wild-Type activity. If there is a previous study highlighting the activity of this mutant, please cite. Otherwise, please present supporting evidence that the activity of the triple mutant is unaffected.

Our apologies for this oversight. We have clarified that the mutant has indistinguishable activity from WT as published previously (pasted below for convenience).

Figure Redacted

Proton efflux is initiated by the addition of increasing concentrations of NaCl/LiCl, and apparent ion-binding affinities for NhaA wild type (closed circle) and mutant (open circle) at pH 8.5 were calculated: $K_M\text{Na}^+$ wild type (mean \pm SD): 1.8 ± 0.2 ; $K_M\text{Na}^+$ mutant: 1.6 ± 0.1 ; $K_M\text{Li}^+$ wild type: 4.1 ± 0.5 ; $K_M\text{Li}^+$ mutant: 4.1 ± 0.3 . (C) pH dependence of NhaA $\text{Na}^+(\text{Li}^+)\text{-H}^+$ antiporter activity for wild type (closed circle) and mutant (open circle) were measured in proteoliposomes by the level of ACMA dequenching as in B at the indicated pH values after the addition of saturating NaCl/LiCl at pH 8.5

2. The authors mention that the specific lipid environment plays a critical role in regulating NHE9*, and that the endosomal lipidic environment is critical to NHE9* activity. However, the MD simulations performed in the manuscript don't use a complex membrane, but rather use a simplified membrane (POPC with the PIP2 lipid embedded) for their simulations. This feels counter-intuitive, and I suppose the results would be better explained by simulations performed in a realistic membrane environment.

Coarse-grain simulations of NHE9*. Coarse-grained molecular dynamics simulations. The NHE9* CC structure were converted to MARTINI3 representation using martinize2 and were embedded into the 90% POPC and 10% bilayer containing POPI(3,5)P2 and solvated in 0.15 M NaCl using insane.py. The bilayer was energy minimised using the steepest descent algorithm, equilibrated for 100 ns and a production run was conducted for 5 us for 4 repeats. All simulations were simulated with a 20 fs timestep using GROMACS 2022.172. The production runs were conducted under 323 K using the v-rescale thermostat. The pressure of all systems was maintained at 1 bar using a Parinello-Rahman barostat

a. Dark blue (lysine residues) and light blue (average position of PI(3,5)P2 lipid). **a.** View of NHE9* homodimer (sand) from above. **b.** Side-view of NHE9* homodimer. The gold spheres show the position of the phosphate beads from the membrane bilayer lipids.

We thank the reviewer for the suggestion. However, given the timescale of all-atoms simulations, we will not be able to get an equilibrated complex bilayer to show how complex lipids modulate the transporter activity unless we place the lipid within the binding site. Indeed, we tried coarse-grained simulation (5 us x 4 repeats) to show that once the dimer is formed, PI(3,5)P2 cannot gain access to the binding site. We have clarified in the manuscript that the β -hairpins are dynamic flexible linkers and so we expect that they adopt a more open position prior to full engagement with the lipid. Unfortunately, the time-scales for these large domain motions make this difficult to model, but this is something we would like to explore in the future.

3. Additionally, the simulations seem too short for the overall system (250 ns), since it has been shown that membrane proteins take a much longer time to equilibrate (~0.5 - 1 microseconds)

We have now extended all of our simulations to 500 ns and update the figures and the conclusions remain the same.

4. There are many interesting experimental observations reported in this study which could have benefited by the physical insights obtained from the MD simulations. However, authors have chosen a narrow question related to mutations in charged residues to obtain an expected results. MD simulations section of the manuscript could have been improved with better design of questions for the computational investigation.

We thank the reviewer for their interest in our studies. We often employ MD simulations with our work, but as you know they take a lot of time to compute and we felt that in the first instance we would use them to better evaluate lipid-bound stability and selectivity between the lipids to support our structure and function studies. However, we would like to explore membrane driven oligomerization in the presence of different lipids, but these are computationally very challenging. Our key finding is not just simply charge mutation studies, but also the selectivity for PI(3,5)P₂ versus PI(4,5)P₂.

Minor comments:

The title could mention NhaA, since the manuscript spends a significant time discussing the NhaA structure.

We have discussed this at some length. While I think a title change would better reflect the overall study, I have already published a review for Nature referring to the NHE9 work that was posted on bioRxiv with the title used in our submission. I also would like to reach the trafficking community and I think they are more likely to pick up the article if the focus is on NHE9 and PIP₂ lipids.

Abstract:

Line 25: ‘all’ cells? Could the authors be more specific?

Yes, we have changed to “every cell membrane”. This family is ancient and is thought that early life acquired Na⁺/H⁺ exchangers to convert natural geothermal H⁺ gradients in deep sea vents into Na⁺ gradients prior to the split between bacteria and eukaryotes. We are not aware of any cell that does not have a Na⁺/H⁺ exchanger. They are as fundamental as K⁺ or Na⁺ channels.

Line 31: dependent on of the lipid cardiolipin. The authors use “horse” throughout the paper. Could they replace it with the scientific name (*E. caballus*)?

Yes, happy too. The name has been changed

Lines 38-41: Break the long sentence into multiple sentences to improve readability.

Yes, this has been corrected, thank you.

Introduction:

Lines 75-76: Citation missing

Marked

Line 90, 93: Citation style changes mid paragraph.

Yes, this has been corrected, thank you.

Results:

Lines 107-114: The authors don’t mention the length of the overall resolved structure (which residues were successfully modelled, etc.)

This has now been added, thank you.

Line 110: Mutation style is inconsistent: (A109T on line 11), while the rest of the manuscript uses three-letter mutations. I would personally recommend to change all three-letter names to single letter, to improve readability.

Thank you. The mutation style has been changed to the single letter amino acid code.

Line 161: Citation missing

Citation was added after the figure call-out and has now been put before it so that its clearer to spot.

Line 165: Extended Data Fig. 4a instead of 3a?

Thank you.

Line 183: Could the authors present the 60° angle in Fig. 2?

Yes, this has now been included.

Line 184: confidently?

Line 186: ECH1 is not labeled in Fig.2

Thank you, this has now been updated.

Line 196: Is NHE9*CC resolved at the same pH as NHE9*? Please clarify.

The purification for NHE9*CC was carried out at pH 8.

Lines 328-329: The RMSD < 3Å does not necessarily imply that the lipid stayed < 3Å from its initial position. Please correct.

This sentence has now been corrected.

What software were used to compute the RMSD, and contact probabilities? Citations are missing. Could the authors use p-values to compare the distributions in Fig. 4b-e?

Line 345: confidently?

You're right, it's a bit unnecessary and ambiguous. The word has been removed.

Line 351: phylogenic or phylogenetic?

It's been corrected, thank you.

Lines 361-362: Could the authors further elaborate on the conclusion? The explanation for the promiscuity of the endosomal transporters seems incomplete.

You're right it's a bit ambiguous here and this is partially because we would like to explore K⁺ vs Na⁺ selectivity in a follow up paper focused on our unpublished NHE6 structure. We can confirm both NHE6 and NHE9 transporters bind K⁺ and we are trying to pinpoint the molecular basis for this selectivity. However, the current study is already fairly large and we decided we would rather answer this more thoroughly in a follow-up paper. We have modified the sentence so it highlights that ion selectivity is unclear.

Line 424: There Their

Changed

Line 627: Wrong citation for CHARMM-GUI, it should be the all-atom citation instead of the coarse-grained

Thank you. This has now been corrected.

- 1 Gupta, K. *et al.* The role of interfacial lipids in stabilizing membrane protein oligomers. *Nature* **541**, 421-424 (2017). <https://doi.org:10.1038/nature20820>
- 2 Rimon, A., Mondal, R., Friedler, A. & Padan, E. Cardiolipin is an Optimal Phospholipid for the Assembly, Stability, and Proper Functionality of the Dimeric Form of NhaA Na(+)/H(+) Antiporter. *Sci Rep* **9**, 17662 (2019). <https://doi.org:10.1038/s41598-019-54198-8>
- 3 Matsuoka, R. *et al.* Structure, mechanism and lipid-mediated remodeling of the mammalian Na(+)/H(+) exchanger NHA2. *Nat Struct Mol Biol* **29**, 108-120 (2022). <https://doi.org:10.1038/s41594-022-00738-2>
- 4 Nakamura, N., Tanaka, S., Teko, Y., Mitsui, K. & Kanazawa, H. Four Na⁺/H⁺ exchanger isoforms are distributed to Golgi and post-Golgi compartments and are involved in organelle pH regulation. *J Biol Chem* **280**, 1561-1572 (2005). <https://doi.org:10.1074/jbc.M410041200>
- 5 Kondapalli, K. C. *et al.* Functional evaluation of autism-associated mutations in NHE9. *Nat Commun* **4**, 2510 (2013). <https://doi.org:10.1038/ncomms3510>
- 6 McCartney, A. J., Zhang, Y. & Weisman, L. S. Phosphatidylinositol 3,5-bisphosphate: low abundance, high significance. *Bioessays* **36**, 52-64 (2014). <https://doi.org:10.1002/bies.201300012>
- 7 Hasegawa, J., Strunk, B. S. & Weisman, L. S. PI5P and PI(3,5)P2: Minor, but Essential Phosphoinositides. *Cell Struct Funct* **42**, 49-60 (2017). <https://doi.org:10.1247/csf.17003>
- 8 Milosavljevic, N. *et al.* The intracellular Na(+)/H(+) exchanger NHE7 effects a Na(+)-coupled, but not K(+)-coupled proton-loading mechanism in endocytosis. *Cell Rep* **7**, 689-696 (2014). <https://doi.org:10.1016/j.celrep.2014.03.054>
- 9 Pedersen, S. F. & Counillon, L. The SLC9A-C Mammalian Na(+)/H(+) Exchanger Family: Molecules, Mechanisms, and Physiology. *Physiol Rev* **99**, 2015-2113 (2019). <https://doi.org:10.1152/physrev.00028.2018>
- 10 Cavet, M. E., Akhter, S., de Medina, F. S., Donowitz, M. & Tse, C. M. Na(+)/H(+) exchangers (NHE1-3) have similar turnover numbers but different percentages on the cell surface. *Am J Physiol* **277**, C1111-1121 (1999). <https://doi.org:10.1152/ajpcell.1999.277.6.C1111>